# WHAT ALGORITHMS CAN TRANSFORMERS LEARN? A STUDY IN LENGTH GENERALIZATION

**Hattie Zhou**[*§†], **Arwen Bradley**[†], **Etai Littwin**[†], **Noam Razin**[‡†], **Omid Saremi**[†],
**Josh Susskind**[†], **Samy Bengio**[†], **Preetum Nakkiran**[†]

[†]Apple  [§]Mila, Université de Montréal  [‡]Tel Aviv University

## ABSTRACT

Large language models exhibit surprising emergent generalization properties, yet also struggle on many simple reasoning tasks such as arithmetic and parity. In this work, we focus on length generalization, and we propose a unifying framework to understand when and how Transformers can be expected to length generalize on a given task. First, we show that there exist algorithmic tasks for which standard decoder-only Transformers trained from scratch naturally exhibit strong length generalization. For these tasks, we leverage the RASP programming language (Weiss et al., 2021) to show that the correct algorithmic solution which solves the task can be represented by a simple Transformer. We thus propose the RASP-Generalization Conjecture: Transformers tend to learn a length-generalizing solution if there exists a short RASP-L program that works for all input lengths. We present empirical evidence to support the correlation between RASP-simplicity and generalization. We leverage our insights to give new scratchpad formats which yield strong length generalization on traditionally hard tasks (such as parity and addition), and we illustrate how scratchpad can hinder generalization when it increases the complexity of the corresponding RASP-L program. Overall, our work provides a novel perspective on the mechanisms of length generalization and the algorithmic capabilities of Transformers.

## 1 INTRODUCTION

Large language models (LLMs) have shown impressive abilities in natural language generation, reading comprehension, code-synthesis, instruction-following, commonsense reasoning, and so on (Brown et al., 2020; Chen et al., 2021; Chowdhery et al., 2022; Lewkowycz et al., 2022; Gunasekar et al., 2023; Touvron et al., 2023). However, when evaluated in controlled studies, Transformers often struggle with systematic generalization (Nogueira et al., 2021; Ontañón et al., 2022; Dziri et al., 2023; Wu et al., 2023; Saparov et al., 2023). It is thus not clear how to reconcile Transformers' seemingly-impressive performance in some settings with their fragility in others.

In this work, we aim to understand the factors that determine a standard decoder-only Transformer's ability to generalize systematically. Recent studies have focused on length generalization on algorithmic tasks as a measure of how well language models can learn to reason (Nogueira et al., 2021; Kim et al., 2021; Anil et al., 2022; Lee et al., 2023; Dziri et al., 2023; Welleck et al., 2022; Liu et al., 2023). Length generalization evaluates the model on problems that are longer (and harder) than seen in the training set, and is used as a proxy for whether the model has learned the correct problem-solving strategy for the given task. There is currently scattered evidence regarding the length generalization capabilities of Transformers. Standard Transformers trained from scratch on addition and other arithmetic tasks exhibit little to no length generalization (Nye et al., 2021; Nogueira et al., 2021; Lee et al., 2023), and even models finetuned from pretrained LLMs struggle on simple algorithmic tasks (Anil et al., 2022). Dziri et al. (2023) propose that Transformers only solve tasks via "analogical pattern matching" instead of learning the true algorithm, and thus will not generalize robustly. On the other hand, length generalization can occur for particular architectural choices and scratchpad formats (Jelassi et al., 2023; Kazemnejad et al., 2023).

---

[*]Work done while interning at Apple.

| RASP-L Program Exists | | No RASP-L Program | |
|---|---|---|---|
| Task Name | Test EM (+10 length) | Task Name | Test EM (+10 length) |
| Count | 100% | | |
| Mode | 100% | Mode (hard) | 0% |
| Unique Copy | 96% | Repeat Copy | 0% |
| Addition (new) | 100% | Addition | 0% |
| Parity (new) | 100% | Parity | 50% |

(a) Transformer length generalization

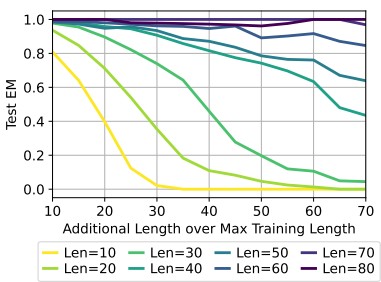

(b) Length generalization on count

Figure 1: **(a)** A selection of tasks studied in this paper partitioned by whether they can be solved by programs in the RASP-L programming language (discussed in Section 4). Test EM denotes the exact match accuracy (EM) for test inputs of length 10 greater than train. We show that all tasks which admit a short solution in the RASP-L programming language also exhibit strong length generalization performance, and vice versa. For certain hard tasks we construct "new" versions which admit RASP-L solutions, by carefully modifying the input and scratchpad format, and these versions length generalize. We also show how poor scratchpad formats can make tasks harder, by giving an example for the Mode task. **(b)** Length generalization for the counting task (described below). Transformers are trained on sequences of varying length, and tested at different levels of out-of-distribution over the maximum training length. Models trained on sequences of length 60 or more exhibit near perfect length generalization up to length 150 (max evaluation length).

As a starting point of our work, we show that there exist algorithmic tasks where Transformers trained from scratch generalize naturally far outside of the training distribution. This observation suggests that length generalization is not inherently problematic for the Transformer architecture, though it clearly does not occur for all tasks. Why then do Transformers exhibit strong length generalization on certain tasks and not others, and what are the mechanisms behind generalization when it occurs? In the following sections, we will propose a unifying framework to predict cases of successful length generalization and describe the possible underlying mechanisms. We discuss additional related works in Appendix A.

**Preview of Length Generalization.** We begin by introducing a simple task that exhibits strong length generalization. The task is "counting": given a prompt `SoS a b >` for numbers $a, b$, the model must count from $a$ to $b$ inclusive, and terminate with "`EoS`". An example is: `SoS 2 5 > 2 3 4 5 EoS` . We train a Transformer with learned positional embeddings on count sequences of lengths up to 50, with random $a$ and $b$ points in $[0, 155]$. This trained model then length-generalizes near perfectly when prompted to count sequences of length 100 (see Figure 1b).

**Possible Mechanisms.** It is helpful to first consider: why should length generalization be possible at all? The crucial observation is that the Transformer architecture is already equipped with a natural notion of length-extension. If we omit positional encodings for simplicity, then a fixed setting of Transformer weights defines a sequence-to-sequence function on sequences of *arbitrary length*. If this function applies the correct transformation for inputs of any length in the context, then we can expect it to length generalize.

For the count task, length generalization is possible if the model somehow learns a correct algorithm to solve the count task. One such algorithm is as follows. To predict the next token:

1. Search for the most-recent `SoS` token, and read the following two numbers as $a, b$.

2. Read the previous token as $x$. If `(x=='>')`, output $a$. If `(x==b)`, output `EoS`.

3. Otherwise, output $(x + 1)$.

This program applies to sequences of all lengths. Thus, if the model ends up learning this program from short sequences, then it will automatically length-generalize to long sequences. The discussion so far could apply to any auto-regressive model, not just Transformers. What is special about the

Transformer architecture and the count task, though, is that a Transformer can *easily represent* the above program, uniformly for all input lengths. This is not trivial: we claim that the same exact Transformer weights which solve the task at length 20, can also solve the task at length 50, and all greater lengths (Figure 1a). In fact, we provide evidence in Appendix C that the trained models are actually implementing this algorithm.

The main message of our work, inspired by the above example, is that it is actually possible for Transformers to approximately[1] learn a length-generalizing algorithm, if the correct algorithm is both possible and "simple" to represent with a Transformer. To reason about what is "simple" for Transformers, we leverage the RASP programming language (Weiss et al., 2021; Lindner et al., 2023), which is essentially an "assembly language for Transformers." (We describe RASP, and our extension of it RASP-L, in Section 4). We then propose the following toy model of learning.

---

**Toy Model (RASP Learning).** *For symbolic tasks, Transformers tend to learn the shortest RASP-L program which fits the training set, if one exists. Thus, if this minimal program exists and correctly length-generalizes, then so will the Transformer.*

---

We find this toy model to be a useful conceptual tool for predicting when Transformers will length-generalize, and is consistent with our experiments on many tasks. Although many prior works have conjectured similar "simplicity bias" in Transformers (Abbe et al., 2023; Bhattamishra et al., 2023), the notion of simplicity we use here is tailor-made for the Transformer architecture: by construction, each line of RASP can be compiled into at most 1 layer of a Transformer. Moreover, this is a notion of simplicity over *programs*, rather than over functions with fixed input dimension. Our main phenomenological conjecture, described in Section 2, is a more precise version of this toy model. On the theoretical side, we also give an example where the "min-degree-interpolator" model of learning from Abbe et al. (2023) does not produce correct predictions for Transformers, but our conjecture does (Appendix H).

The RASP perspective thus helps unify our understanding of length generalization: instead of developing specialized tools to study generalization of each individual task (such as addition, multiplication, parity, with and without scratchpads, etc), we can now apply a generic hammer to all tasks. To predict whether a task is likely to generalize, the RASP Conjecture tells us we should first see if it can be solved by a RASP-L program for all input distributions. While we focus on studying length generalization in this work, we expect the intuitions developed here can apply to systematic generalization more broadly.

## 2 MAIN CONJECTURE

We now describe our main conjecture. In this conjecture, and throughout the paper, we consider Transformers "trained to completion," meaning trained to near-optimal performance on their training distribution. That is, we assume that in-distribution generalization is achieved nearly-optimally, and focus our attention on the induced out-of-distribution generalization. The exact training procedure we consider is given in Section 3.

A key ingredient we develop is a restriction of RASP which we call RASP-L, that we describe in more detail in Section 4. For now, it suffices to say RASP is a human-readable programming language which defines sequence-to-sequence programs that can be compiled into Transformer weights, such that each line of RASP roughly corresponds to one Transformer layer. Our conjecture follows.

**RASP-Generalization Conjecture.** *A decoder-only autoregressive Transformer is likely to length-generalize when trained to completion on an algorithmic task if the following conditions hold.*

1. ***Realizability.*** *The true next-token function for the task can be represented by a single decoder-only Transformer which works on all input lengths.*

2. ***Simplicity.*** *This representation is "simple", meaning it can be written in RASP-L (a learnable subset of RASP defined in Section 4).*

---

[1]We do not claim the model will be exactly equivalent to the correct algorithm on all possible inputs; this is unlikeley for technical reasons— numerical precision, noise in the training and sampling procedures, finite optimization time, etc. However, we can hope for strong approximations.

> 3. ***Diversity.*** *The training data is sufficiently diverse, such that there does not exist any shorter RASP-L program which agrees with the task in-distribution but not out-of-distribution.*

The first condition of realizability is actually quite stringent, because it requires a *single Transformer* to be able to solve the task at *all lengths*[2]. Decoder-only Transformers (also known as causal Transformers) define a particular computational model, and not all sequence-to-sequence tasks can be solved within this model. For example, next-token functions which require $\Omega(n^3)$ computation time on inputs of length $n$ provably cannot be represented by a Transformer, however large— a consequence of the Time Hierarchy Theorem (e.g. Arora & Barak (2009)). The realizability condition may seem stronger than required, because in practice, we do not actually need length generalization for arbitrary unbounded lengths— only lengths up to some maximum context size. Nevertheless, we find that considering representability in the unbounded length setting is a good heuristic for learnability in bounded length settings. Intuitively, if a task requires a different Transformer for each input length, then it may be an "unnatural" task for Transformers, and unlikely to generalize well.

We emphasize that our conjecture is primarily *phenomenological*, as opposed to *mechanistic*. That is, we do not make strong claims about which exact function the Transformer learns, and whether or not it is equivalent to a RASP-L program. Our main conjecture characterizes *which* tasks Transformers are likely to length-generalize on, and not *why* or *how* they do so. Although we believe the toy model is a plausible mechanism that implies our conjecture, we leave investigating this more fully as an important question for future work.

**Limitations of Scope and Strength.** We acknowledge that our Main Conjecture is not fully formal, because there are aspects we do not fully understand. For example, we cannot precisely predict the *extent* of length generalization for different tasks. Moreover, since it is likely intractable to determine the minimum RASP-L program that fits a given training set, we cannot predict a priori what forms of "data diversity" are required to ensure strong length generalization, even if our conjecture holds true. Nonetheless, we view our conjecture as a step forward in understanding the implicit bias of Transformers, as it has more predictive power than many prior theories. For example, in Appendix H we give a simple theoretical setting where the popular "min-degree-interpolator" model of learning from Abbe et al. (2023) does not correctly predict Transformers' out-of-distribution behavior, but our conjecture does. Developing more formal and precise conjectures is an important question for future work.

## 3 EXPERIMENTAL SETUP

A *Transformer* (Vaswani et al., 2017) refers to a decoder-only causal Transformer architecture with constant depth, width, and fixed setting of weights, along with any computable positional embedding scheme[3]. As a technical point, we allow the transformer weights to take values in the extended real line $\mathbb{R} \cup \{\pm\infty\}$, to allow saturating the softmax at arbitrary context lengths[4]. We consider only greedy sampling throughout, since our tasks are deterministic.

We train all of our models to convergence on the train distribution where possible. For all tasks, the length of training examples is sampled uniformly from length 1 up to the max training length. We train Transformer models from scratch and use learned positional embedding on all tasks. At train time, we "pack the context", filling the Transformer's context window with multiple independent samples of the task, and we randomly shift the Transformer along its context window. This procedure of packing and shifting the context mirrors standard practice in LLM training on real data (Karpathy, 2023; Brown et al., 2020), but is typically not done in prior works using synthetic tasks[5]. It is an important detail: packing and shifting the context allows all positional embeddings to be trained, and encourages the transformer to treat all positions symmetrically. At test time, we

---

[2] This corresponds to a *uniform* model of computation, in the terminology of computational complexity. See Merrill & Sabharwal (2023) for a discussion of uniformity in the context of Transformers.

[3] This is a technical detail: we consider position encoding schemes which can be uniformly generated, i.e. there exists a Turing machine which on input $(i, n)$, produces the positional embedding vector for index $i$ out of $n$ total.

[4] This bypasses the limitations presented in Hahn (2020), which exploit non-saturating softmaxes.

[5] One exception to this is Liu et al. (2023), which performed a similar type of shifting.

evaluate examples without packing and shifting. We measure the exact match (EM) on all outputs, which is 1 if the entire output sequence is correct, and 0 otherwise. Full details in Appendix B.

# 4 RASP-L: WHAT ALGORITHMS CAN TRANSFORMERS LEARN?

We will now define a version of the RASP programming language (Weiss et al., 2021), which we call RASP-L, to heuristically capture the set of algorithms which Transformers can both represent and learn. To apply our conjecture, we need to determine if a given task can be solved in RASP-L. Since RASP-L programming is fairly non-intuitive, we present a "standard library" of useful RASP-L functions, which one can compose to solve more complex tasks. We also discuss which types of operations are hard or impossible in RASP-L, to give intuition about which programs cannot be representable or learnable by Transformers.

## 4.1 RASP: A PRIMER

The original RASP language can be thought of as a domain-specific-language for specifying Transformer weights, in human-readable form (Weiss et al., 2021). Importantly, RASP was designed for the computational model of Transformers, so short RASP programs define functions which are "easy to represent" for Transformers. Although RASP was conceived as a separate language with its own syntax, we can also realize RASP as a restricted subset of Python where only a few operations are allowed. We show how to do this explicitly in Appendix F.1, and just include a few examples here. Every RASP program accepts an input sequence of length $n$, for all $n \in \mathbb{N}$, and returns an output sequence of the exact same length— just like a Transformer. The core operations allowed in RASP are: arbitrary elementwise operations over sequences (`map` and `seq_map`), and a very particular type of non-elementwise operation `kqv`, which simulates a causal Attention layer. Moreover, no control flow is allowed; all programs must be straight-line programs, with no branching or loops.

For example, suppose we want to write a causal RASP program which always outputs the second-to-last token of the input sequence, e.g. in Python: `def f(x): return x[-2]`. To represent this function in a causal sequence-to-sequence manner, we need to take in the sequence `x` and output the same sequence but shifted by 2. In pure Python, the function is `lambda x: [0]*2 + x[2:]`, where we pad the shifted `x` with 2 tokens in the front to maintain the same dimension. This lambda function can be implemented in RASP as: `lambda x: kqv(indices(x)+2, indices(x), x, equals)`. This RASP program uses the fact that positional embeddings can be specially constructed such that shift-by-two is an *elementwise* operation on these embeddings. Then, `kqv` function simulates an attention layer where the current position indices (as query) are matched to the shifted-by-2 indices (as key), and applied to input sequence `x` (as values).

Crucially, since we study autoregressive decoder-only Transformers, we must use the *causal* version of RASP, where all seq-to-seq operations are executed causally. Moreover, while RASP programs define sequence-to-sequence functions, we interpret them as sequence-to-token functions, by taking the last token of output sequence as the next-token prediction (in the standard autoregressive manner). This setting differs from most prior literature on RASP, which typically consider non-causal models, and these differences significantly change the nature of RASP programming.

**Intuition.** The intuition to takeaway from this example is that RASP only allows parallelizable operations, because Transformers are an inherently *parallel* model of computation. This makes performing inherently-sequential computation, such as iterating through each input symbol and updating an internal state, tricky if not impossible to write in RASP. This is why loops are not allowed in RASP: because a Transformer has only constant depth, and cannot directly simulate an arbitrary number of loop iterations. The one way of bypassing this limitation is to exploit the *autoregressive* inference procedure: since the model is called iteratively at inference time, this effectively provides an "outer-loop" that can enable sequential computation which uses the input context as the state. This is exactly what scratchpads enable, as we elaborate in Section 5.

## 4.2 RASP-L: LEARNABLE RASP

The original RASP technically allows for certain operations which are possible to represent, but not "easy" to represent or learn. For example, arbitrarily-complex tokenwise operations $\mathbb{R} \rightarrow \mathbb{R}$

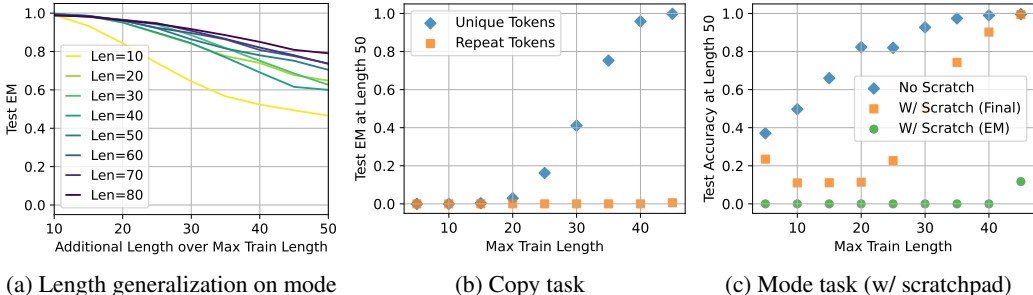

(a) Length generalization on mode  (b) Copy task  (c) Mode task (w/ scratchpad)

Figure 2: **(a)** Length generalization performance for the mode task. All models generalize perfectly to 10 or more additional tokens at test time. **(b)** Performance on copy tasks for test sequences of length 50, and varying train lengths. Copying unique tokens makes length generalization much easier, since it can be solved by an induction head. **(c)** Performance on mode task for test sequences of length 50, and varying training lengths. The scratchpad hurts in this setting (both final answer accuracy and exact match), since it is not computable by a short RASP-L program.

are allowed. To disallow such pathologies, we define a "learnable" subset of RASP which we call RASP-L. The technical details of RASP-L are in Appendix F.2, but the primary restrictions are: all variables are bounded integers (`int8`) to avoid arbitrary-precision and numerical stability issues, and token indices are treated specially. Roughly, token indices in RASP-L can only be operated on in simple ways (order comparison, predecessor, successor)— arbitrary arithmetic involving indices is not allowed. We find that empirically, such operations involving index-arithmetic are often not learned robustly, as we elaborate on in Appendix F.2.

**A RASP-L Standard Library.** In Appendix F.3.2 we provide a small library of useful functionality built on the RASP-L core. This also serves as a representative sample of functions that are "easy" to implement RASP-L, to build intuition about programming causal Transformers. The library includes `where`, which is functionally equivalent to `numpy.where`, and allows for rudimentary branching (though inefficiently, since all branches are always evaluated). It also includes the fairly self-explanatory `maximum`, `minimum`, `argmin` and `argmax`. Recall that all functions are seq-to-seq and causal, so for instance `maximum` returns the array of running maxima. The functions `num_prev`, `firsts`, and `induct` all take an input sequence x and a query sequence q, and return, respectively, the number of previous elements of x equal to each `q[i]`, the index of the first occurrence of each `q[i]` in x, and the token *following* the first occurrence of `q[i]` in x.

## 4.3 CASE STUDIES

In this section, we experimentally evaluate 3 tasks that have simple RASP-L programs and 3 tasks that do not. We show that RASP-L representability is correlated with length generalization performance. The three easy tasks we consider are: count, mode, and copy with unique tokens. We provide the RASP program for these tasks in Appendix F.3 (Listings 3, 4, and 5, respectively). Detailed training procedures and hyperparameters are provided in Appendix B.

**Count.** We described the count task in the Introduction and showed results in Figure 1b. This task can be solved by a RASP-L program that essentially translates the pseudocode in the Introduction (details in Listing 3). We find that models trained on count can generalize near perfectly to double the training lengths. It is crucial that our training distribution contain samples ranging from length 1 to maximum training length, which adds diversity as we scale up training length. These factors are necessary in preventing shortcut programs from being learned: no generalization is observed if we train on sequences of all the same length.

**Mode.** The mode task identifies the most frequent element in a sequence. We constrain the sequences such that the answer is unique. An example is: | a | b | b | c | b | a | c | b | > | b |. Figure 2a shows the results on mode, when training and testing on random sequences from an alphabet of 52

symbols. We find that models trained on mode generalizes strongly to sequence lengths much longer than the maximum training lengths. Interestingly, increasing training set complexity here does not result in huge improvements in length generalization: Even models trained on max sequence length of only 10 can achieve a median test accuracy of 50% of sequences of length 60.

**Copy with unique tokens.** The copy task repeats the prompt sequence in the output. We constrain the sequences to have all unique tokens in the prompt. An example is: `a c d b > a c d b`. Figure 2b shows the results on copy with unique tokens. For models trained on sequence length up to 40, we find that they can generalize perfectly to length of 50. Intuitively, this task is easy because we can leverage something called an "induction head" (Olsson et al., 2022). Induction heads work by identifying a previous instance of the current token, find the token that came after it, and predict the same completion to the current token. Olsson et al. (2022) found that induction heads are reliably learned even by simple Transformers, and conjectured them to be a component of what enables in-context learning in LLMs. Induction heads are simple to implement in RASP-L, as the `induct` function. Thus, the next token can be generated by simply using an induction head on the current token, since all tokens are unique. This is exactly what the RASP-L program does, in Listing 5.

Next, we identify three tasks that do not admit simple RASP-L solutions: addition, parity, and copy with repeating tokens. We discuss reasons why Transformer models struggle to generalize on these tasks by highlighting the operations these algorithms require, but which are unnatural for a Transformer to represent.

**Addition & Parity.** Both these tasks have been studied as difficult tasks for Transformers: models trained from scratch show little to no length generalization on addition (Nye et al., 2021; Lee et al., 2023) and parity (Bhattamishra et al., 2020; Chiang & Cholak, 2022; Ruoss et al., 2023; Delétang et al., 2023), and even pretrained LLMs cannot solve these tasks robustly (Brown et al., 2020; Chowdhery et al., 2022; Anil et al., 2022) without careful prompting (Zhou et al., 2022b).

Indeed, addition is also difficult to write in RASP-L. To see why, consider the standard addition program shown in Appendix F.3.6. This algorithm requires the carry value to be propagated in reverse order from least- to most-significant digit, but this is difficult to simulate due to causal masking. Moreover, the most prohibitive aspect is the index-related operations. The standard addition algorithm requires index-arithmetic (e.g. finding the middle of the prompt sequence) and precise indexing operations (e.g. look up the corresponding summand digits for the current output digit). Such operations are forbidden in RASP-L, as they require index-arithmetic which are difficult to represent in a global, length-generalizing way (see Appendix F.2 for more discussion). Similarly, parity without any scratchpad requires operations that are forbidden under RASP-L. Since a Transformer cannot update its state sequentially, it must solve parity with parallel operations only. Intuitively, this requires taking the sum of the entire sequence, then determining the parity of the sum. This cannot naturally be computed in a numerically stable way for arbitrarily large sums, and we cannot expect to learn a 'sum' operation which generalizes to numbers larger than the training sequences. Indeed, many works have shown that a Transformer cannot even fit the training set of parity sequences over some minimal length (Bhattamishra et al., 2020; Chiang & Cholak, 2022). Under our experimental settings, we find that *no* length generalization is observed for both addition and parity tasks. We evaluate different test lengths in increments of 5, and all 20 runs on all training lengths up to 45 showed a test EM of approximately 0% on addition and parity.

**Copy with repeating tokens.** For this task, we constrain the sequences to consist only of 2 possible tokens. An example is: `a a b a > a a b a`. Since the tokens are no longer unique, the induction-head is no longer helpful. Instead, the model must perform precise index-arithmetic, which is difficult for the same reason that indexing is difficult in addition. We show in Figure 2b that models completely fail to generalize to longer lengths on this task.

## 5 APPLICATION: IMPROVING LENGTH GENERALIZATION

In this section, we demonstrate how our RASP conjecture can go beyond post-hoc explanations, by constructing interventions that predictably change length generalization performance. We study how reformatting tasks to allow shorter RASP-L programs can improve generalization performance, and how increasing diversity in the training data allows the model to perform well on tasks that require more complex RASP-L programs.

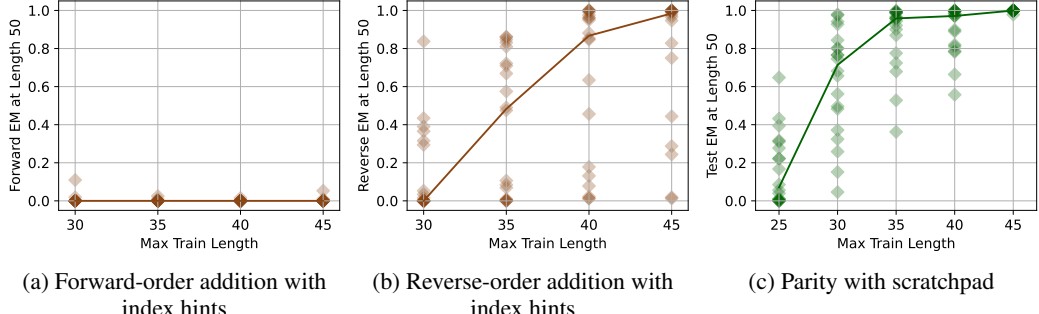

(a) Forward-order addition with index hints      (b) Reverse-order addition with index hints      (c) Parity with scratchpad

Figure 3: **Length generalization on addition and parity.** Plot shows 20 individual trials per length, as well as their median (solid line). **(a)** Shows generalization performance for forward addition with index hints on hard carry examples of length 50. No length generalization is observed. **(b)** Shows the generalization of reverse addition with index hints on hard carry examples of length 50. Most runs start to length generalize perfectly on 50 digit addition once the training length is greater than 40. **(c)** Shows generalization performance for parity with scratchpad, on length 50 inputs. Most runs start to generalize perfectly on 50 digit addition once the training length is greater than 35.

## 5.1 DEEP DIVE ON ADDITION

**Reducing the RASP-L complexity for addition.** In the previous section, we noted two aspects of a naive addition algorithm that pose problems for RASP-L: index-arithmetic (to query the summand digits for the current output digit), and non-causality (for the carry). To address the difficulty with indexing operations, we can leverage induction heads to simplify the addition algorithm for a Transformer by adding "index hints" to the prompt and answer: For example, $\boxed{5}\boxed{4}\boxed{+}\boxed{3}\boxed{7}\boxed{>}\boxed{9}\boxed{1}$ becomes $\boxed{a}\boxed{5}\boxed{b}\boxed{4}\boxed{+}\boxed{a}\boxed{3}\boxed{b}\boxed{7}\boxed{>}\boxed{a}\boxed{9}\boxed{b}\boxed{1}$. This enables us to get the corresponding digits for each sum step by calling `induct` on its index hint (`a` or `b`), thus sidestepping the need to precisely access and manipulate positional indices. To address non-causality of the carry operation, we can format the output in reverse-order. For example, $\boxed{a}\boxed{5}\boxed{b}\boxed{4}\boxed{+}\boxed{a}\boxed{3}\boxed{b}\boxed{7}\boxed{>}\boxed{a}\boxed{9}\boxed{b}\boxed{1}$ becomes $\boxed{a}\boxed{5}\boxed{b}\boxed{4}\boxed{+}\boxed{a}\boxed{3}\boxed{b}\boxed{7}\boxed{>}\boxed{b}\boxed{1}\boxed{a}\boxed{9}$. This enables simple and causal propagation of the carry where each step can reference the previous output to determine the current carry, similar to the standard addition algorithm. A RASP-L program for reverse-order addition, with index-hints, is provided in Listing 7. Although reversing the answer digits greatly simplifies the carrying procedure, it is still possible to implement an algorithm for addition in forward order (Listing 8). This algorithm is nontrivial, because we essentially need to propagate a carry through all $n$ input digits in order to determine the first digit of output, the most significant digit (see Figure 12). We show how to construct a RASP-L program for forward-order addition in Listing 8. Comparing Listing 7 and Listing 8 reveals how much more complicated the forward algorithm is— it results in a much longer RASP-L program.

**Index hints enables generalization on addition.** We evaluate addition in two settings: "easy" carry and "hard" carry. In easy carry, the two numbers are sampled randomly and independently— this is what is typically done in the literature. However, uniformly random summands will only produce addition instances with short carry-chains (in expectation)— and for such instances, each output digit only depends on a small number of input digits. We thus also test "hard" carry instances, where we constrain the examples to have the longest possible carry chain for the given length. For example, a hard carry instance of length 3 is $381+619 = 1000$, which requires the model to compute the carry over a chain of 3 digit positions. The performance on "easy" carry is shown in Figure 8, and the performance on "hard" carry in Figure 3. We find that index hints allow both forward and reverse addition to length generalize on "easy" carry. However, on "hard" carry questions that involve carry-chains longer than seen at training, reverse addition maintains strong length generalization while forward addition exhibits no generalization.

**Diversity enables generalization on forward addition.** Another lever for performance improvement suggested by the RASP conjecture is to increase training data diversity, such that shortcut programs can no longer fit the training set. Since forward addition does admit a RASP-L program,

albeit a more complex one, we would expect it is possible to learn if we "try harder," e.g. use a more careful and diverse train distribution. We explore this by training with balanced carry sampling—instead of sampling the two numbers independently, we first sample the length of the carry chain uniformly between 0 and question length, then sample a random question that contains the given carry chain length. This ensures that the model sees a significant percentage of questions containing long carry chains, thus increasing the diversity and difficulty of the training data. The results of the balanced carry training approach for both forward and reverse addition are shown in Figure 9. We see that this more careful training unlocks the model's ability to length generalize on forward addition, even under the hard carry evaluation. To our knowledge, these results demonstrate the first instance of strong length generalization on decimal addition for Transformers trained from scratch.

## 5.2 WHY DO SCRATCHPADS HELP?

The RASP conjecture provides a natural way to understand why scratchpads (Nye et al., 2021; Wei et al., 2022) can be helpful: scratchpads can simplify the next-token prediction task, making it amenable to a short RASP-L program. One especially common type of simplification is when a scratchpad is used to "unroll" a loop, turning a next-token problem that requires $n$ sequential steps into $n$ next-token problems that are each only one step. In the following examples, we construct "good" scratchpads which simplifies the target RASP-L program, and "bad" scratchpads which seem natural to humans but require a more complex RASP-L program than the original task. We show that these interventions lead to predictable changes in length generalization.

**Scratchpad enables generalization on parity.** We leverage the "unrolling" intuition to design a scratchpad for parity. Similar to addition, we add index hints to the prompt to simplify the indexing operation. In the scratchpad output, we locate index hints that precede each 1 in the prompt, and keep track of the running parity with symbols $+$ (even) and $-$ (odd). The last output token corresponds to the final answer. For example: `a0b0c1d1e0>+c-d+`. Figure 3c shows the exact match performance of the proposed parity scratchpad. We see that some of the runs trained with sequences up to 30 in length can generalize perfectly on sequences of length 50. When training length reaches 45, all models achieve perfect length generalization on length 50. These results demonstrate the *first* instance of strong length generalization on parity for Transformer models trained from scratch.

**Scratchpad *hurts* generalization on mode.** Now we consider the mode task and look at how scratchpad might affect a task that a Transformer is naturally amenable to. A natural algorithm one might come up with is to calculate the counts of each unique token in the sequence, then output the token with the maximum count. To encourage the model to learn this algorithm, we might utilize the following scratchpad, where we output the frequency of each token in ascending order: `abbcbacb>2a2c4bb`. The last token in the scratchpad is then the correct answer. However, although this scratchpad provides more supervision for what algorithm the model should learn, it is a more difficult task when considered in terms of RASP-L. Finding and comparing the frequency is simple to do internally, but converting this implicit representation into an integer token adds additional complexity. We show in Figure 2c that the scratchpad performs significantly worse than no scratchpad, both when measured on exact match and also on the accuracy of the final answer.

## 6 CONCLUSION

We proposed a model for understanding and predicting when Transformers are likely to exhibit strong length generalization on algorithmic tasks. We do so by reasoning about the unique information-flow constraints of the Transformer architecture through a variant of the RASP language. We conjecture that algorithms which are simple to represent by a Transformer are also more likely to be learned, and use this model to predict and improve length generalization on a set of algorithmic tasks. Our toy model demystifies certain observations of strong reasoning and out-of-distribution abilities of Transformers, and shows that they can occur for potentially simple reasons. Although our studies focused on length generalization, the intuitions are not specific to it, and they lay out a path towards a possible mechanism for systematic generalization. Studying these mechanisms beyond length generalization, and their interaction in multi-task settings, are important directions for future work.

ACKNOWLEDGEMENTS

We thank (alphabetically) Samira Abnar, Madhu Advani, Jarosław Błasiok, Stefano Cosentino, Laurent Dinh, Fartash Faghri, Spencer Frei, Yejin Huh, Vaishaal Shankar, Vimal Thilak, Russ Webb, Jason Yosinski, and Fred Zhang for feedback on early drafts and discussions throughout the project.

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

# A    ADDITIONAL RELATED WORKS

Our paper is related to the line of work that seeks to understand the capabilities and limitations of Transformer models when it comes to algorithmic reasoning (Kaiser & Sutskever, 2015; Veličković & Blundell, 2021). Specifically, we focus on simple tasks like arithmetic and study length generalization on the standard Transformer architecture. Related to this, Lee et al. (2023) study how well transformers trained from scratch can learn simple arithmetic tasks, and finds that no length generalization is observed. Nogueira et al. (2021) find that partial length generalization on addition is observed only when models reach 3B parameters and when the addition questions are presented in reverse order. Jelassi et al. (2023) study models trained on addition and find strong generalization performance when using a few examples of longer sequences. However, they required non-standard architectures (non-causal Transformers) and training procedures (artificial padding) to observe this, and still find that the model does not generalize to unseen length that is in between the minimum and maximum lengths seen during training.

Moreover, our work contributes to the study of what makes for effective scratchpads. Other papers have also found that using positional tokens in the prompt can help with length generalization (Nogueira et al., 2021; Li & McClelland, 2023). However, these works do not provide a framework for understanding why these tricks are helpful. A number of papers also study how chain-of-thought style prompting helps with reasoning performance (Wei et al., 2022; Zhou et al., 2022b;a; Creswell & Shanahan, 2022; Madaan & Yazdanbakhsh, 2022), but these focus on in-context learning and do not study the effect of training models on these formats.

Other papers also aim to understand the limits of what Transformers can learn and represent. Bhattamishra et al. (2020) and Delétang et al. (2023) study the ability of Transformers to represent and generalize on families of formal languages. Delétang et al. (2023) evaluated Transformers on tasks at all levels of the Chomsky hierarchy, and found that "Transformers and LSTMs are not well-aligned with the Chomsky hierarchy." This is consistent with our results, since the Transformer computational model does not directly correspond to a natural level in the hierarchy. Valvoda et al. (2022) benchmarked Transformers on randomly-generated tasks produced by deterministic finite-state transducers, and evaluated an even stronger metric than length-generalization (compositional generalization). They found that Transformers only generalize on certain tasks, and only when the training data has sufficient "coverage" of the task. As an aside, we chose not to use randomly-generated tasks in our setting, because our target program space is much larger than Valvoda et al. (2022), so our experimental results would be depend very strongly on the exact sampling procedure for random RASP-L programs. For example, a naive "random RASP-L program" is likely to always produce the constant 0 output, due to degeneracies. Zhang et al. (2023) evaluate the ability of transformer models to emulate the behavior of structurally recursive functions from input-output examples. Liu et al. (2023) study how shallow Transformers can simulate recurrent dynamics representable by finite-state automata. Both works identify shortcut solutions that become brittle on out-of-distribution samples. To address these shortcomings, various works have proposed modifications or alternatives to the Transformer architecture in order to improve generalization, such as Press et al. (2022); Chi et al. (2023); Mahdavi et al. (2022).

Bhattamishra et al. (2023) suggest that Transformer models have an inductive bias towards learning functions with low sensitivity, such as sparse boolean functions, but focus on the in-distribution setting. Abbe et al. (2023) also propose a simplicity bias in Transformers, but use "minimum-degree" as their notion of function simplicity. However, they only consider functions with fixed input dimension rather than programs on arbitrary input lengths.

Lastly, there have been many other approaches to improving length generalization in Transformers. These include studying how various training hyperparameters and design choices influence compositional generalization (Furrer et al., 2021; Ontañón et al., 2022), and designing better positional embeddings (Press et al., 2022; Ontañón et al., 2022; Kazemnejad et al., 2023; Ruoss et al., 2023).

# B    ADDITIONAL EXPERIMENTAL DETAILS

For all experiments, we tokenize every character individually, including digits of a number. We train in the online setting, where each batch is sampled iid from the train distribution instead of from a finite train set — this avoids overfitting issues, and is closer to the training of modern LLMs.

Unless otherwise specified, we evaluate test performance on $5\times$ the batch size number of samples. Unless otherwise specified, we run each experiment on 20 random seeds and report the median of the runs. We select hyperparameters for each task based on what is required to fit the training set. Hyperparameter details can be found in Table 1.

**Count.** For the count task, we train with an alphabet size of 155 and evaluate on test sequences up to 150 in length. Given the nature of the task, we enumerate all possible sequences of each test length at evaluation time. The alphabet is ordered such that the sequence between any two tokens in the alphabet is clearly defined. At train time, the length of each example is sampled uniformly between 1 and the maximum training length.

**Mode.** For the count task, we train on an alphabet of 52 tokens. Each example consist of 5 unique tokens sampled randomly from the alphabet. The length of each example is sampled uniformly between 1 and the maximum training length, and the sequence is randomly sampled from the 5 selected tokens. If there is a tie for the most frequent element, we randomly select a token from one set and changes it to a token from the other set, thus ensuring that there is one unique answer.

**Copy.** For the copy task with unique tokens, we train on an alphabet size of 100. The length of each example is sampled uniformly between 1 and the maximum training length, and the sequence is randomly sampled from the alphabet without replacement. For the copy task with repeat tokens, we use the same sampling procedure, but now on an alphabet size of 2.

**Addition.** For the addition task, we sample the length of each of the two numbers independently, from 1 up to the maximum training length. We then pad the two numbers with 0 in the front such that they have the same length. We pad the numbers with an extra 0 to allow for the potential of an extra digit in the answer due to carry. For addition with index hints, we sample the index hints as a random slice from a longer contiguous block of tokens, to encourage learning all hints and their linear ordering. This is similar to our training strategy for the count task.

**Parity.** For the parity task, we sample the length of each parity sequence from 1 up to the maximum training length. We then sample randomly from $\{1, 0\}$ a sequence of the given length. We note that the definition of length we use is based on the sequence length and not based on the number of 1s in the sequence.

Table 1: Experimental hyperparameters. All experiments use AdamW optimizer and cosine learning rate schedule. Count and Copy use weight decay of 0.1 and grad clip of 0. Parity and Mode use weight decay of 0.1 and grad clip of 1. Addition uses weight decay of 0 and grad clip of 1.

| Task | Model Size | Train Iter | Context Len | Batch Size | Learning Rate |
|------|-----------|-----------|------------|-----------|--------------|
| Count | 6 layer; 8 head; 64 emb | 10000 | 256 | 128 | 1e-3 to 1e-5 |
| Mode | 6 layer; 8 head; 512 emb | 10000 | 256 | 128 | 1e-3 to 1e-6 |
| Copy | 6 layer; 8 head; 512 emb | 100000 | 512 | 128 | 1e-4 to 1e-6 |
| Addition | 6 layer; 8 head; 512 emb | 30000 | 512 | 64 | 1e-4 to 0 |
| Parity | 6 layer; 8 head; 512 emb | 10000 | 512 | 256 | 1e-3 to 1e-6 |

## C  APPENDIX: COUNTERFACTUAL ANALYSIS ON COUNT

In this section, we probe whether models trained on count actually learn the count algorithm that we intuitively want. To reiterate, one simple algorithm that solves the count task is as follows. To predict the next token:

1. Search for the most-recent `SoS` token, and read the following two numbers as $a, b$.

2. Read the previous token as $x$. If `(x=='>')`, output $a$. If `(x==b)`, output `EoS`.

3. Otherwise, output $(x + 1)$.

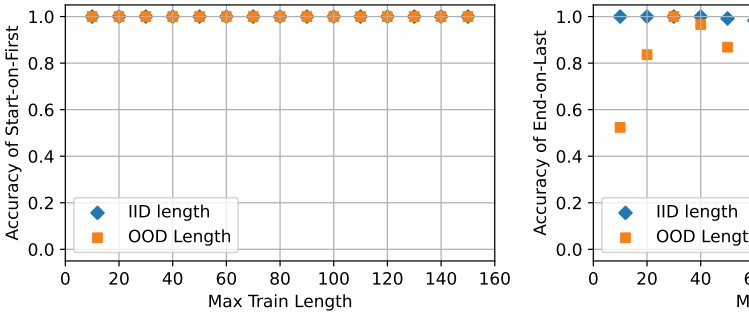
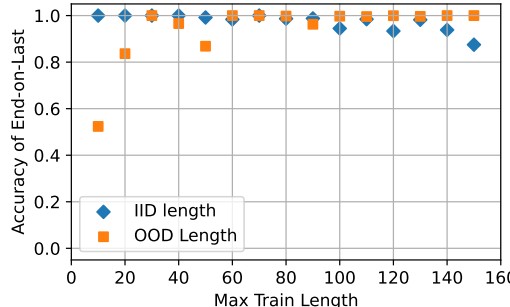

(a) Counterfactual test of starting on first token

(b) Counterfactual test of ending on last token

Figure 4: We measure performance of models trained on the count task on counterfactual tests designed to evaluate whether the model simulates the correct generalizing algorithm on random sequences far out-of-distribution. We see that models always output the start token as the first output, but do not always output the EoS token once the ending token has been outputted.

Since there is no easy way to formally check if the Transformer model learns this exact algorithm internally, we employ the simple heuristic of running the model on counterfactual examples. The allows us to stress test the models behavior and see if it performs the expected algorithmic steps in an input-agnostic way. To do so, we performance inference on randomly generated input sequences that are designed to test the model in four ways:

1. The model should always output the start token in the prompt ($a$) as the first output. (Figure 4a)

2. The model should always output EoS following a token that matches the ending token in the prompt ($b$). (Figure 4b)

3. In all other settings, the model should increment the previous token by 1. (Figure 5a)

4. The model should not output EoS prematurely. (Figure 5b)

We create the counterfactual dataset by sampling start and end tokens of varying distances, then generate a sequence of random tokens of the length specified by the distance between the start and end token. We then pass this sequence through a trained model and look at its predictions at each token position. The goal of the four proposed tests on random sequences is to probe whether the model learned the expected algorithmic behavior rather than learning something that would strongly depend on statistics of the training distribution. We sample examples for in-distribution lengths and out-of-distribution lengths based on the training distribution of each model. For simplicity, we choose 1 model with strong length generalization performance from each maximum training length setting. The performance on each test is shown in Figure 4 and Figure 5.

We see that for the start-on-first test and the increment-by-1 test, all models exhibit near perfect performance both in- and out-of-distribution. For the end-on-last test, we see that models trained with shorter lengths do not learn to robustly output EoS on long test sequences once the ending condition is met. However, on models trained on longer sequences (and has better length generalization), this behavior is more robust. Lastly, when we measure the percentage of EoS which are correct, we see that models that do not have strong generalization also fails to output EoS only at the appropriate time. This failing is observed on both in-distribution and out-of-distribution lengths. This suggests that the failures of length generalization can be attributed to prematurely outputting an EoS token before the full length of the sequence is outputted. Overall, we observe strong correspondence between the model's behavior and what we would expect from the correct algorithm. This lends credence to the intuition that the model learns the correct RASP-L program and generalizes because of it.

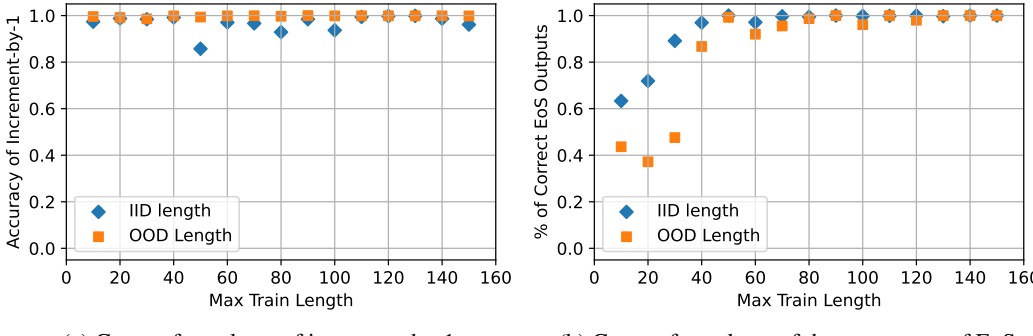

(a) Counterfactual test of increment-by-1

(b) Counterfactual test of the percentage of EoS outputs which are correct

Figure 5: We measure performance of models trained on the count task on counterfactual tests designed to evaluate whether the model simulates the correct generalizing algorithm on random sequences far out-of-distribution. We see that models almost always increments the previous token by 1, no matter what the previous sequence is. However, it sometimes output the EoS token prematurely, especially on lengths longer than seen in training. This likely explains failures of length generalization observed in Figure 1b.

## D  APPENDIX: TRAINING SPEED

The RASP Generalization Conjecture suggests that simple-to-represent programs are also more likely to be learned. Intuitively, we should expect that simple-to-represent programs also exhibits faster optimization during training. In this section, we evaluate different strategies of presenting the addition and the parity task, and see whether the training speed of these variants correspond to the simplicity of their corresponding RASP-L programs (and by extension their length generalization performance).

As discussed in Section 5, reverse addition simplifies the function that the model needs to learn when compared to forward addition. Figure 6a shows the training curves for each of these settings. Consistent with intuition, we see that reverse addition converges more quickly than forward addition.

For the parity task, we introduce an additional scratchpad format for comparison. This scratchpad outputs the sum-mod-10 of the parity sequence before outputting the final parity. An example is `0 0 1 1 0 > 2 , 0`. This scratchpad does not simplify the problem as much as the main scratchpad presented in Section 5 because it does not leverage the autoregressive inference to process the task sequentially. However, it is still simpler than parity without any scratchpad because it helps to simplify the final operation of getting the parity of the sum. Instead of doing this internally, the model can now reference the output of the sum-mod-10 and learn a simple mapping between that and the corresponding parity. Figure 6a shows the training curves for each of these settings. We see that the main scratchpad ("Easy Scratchpad") optimizes much more quickly than the sum-mod-10 scratchpad ("Hard Scratchpad"). We also observe that Easy Scratchpad exhibits significantly stronger length generalization than Hard Scratchpad, shown in Figure 7b. Both scratchpads optimizes much better than parity with no scratchpad, which is unable to even fit the training set and demonstrates no length generalization.

## E  APPENDIX: ADDITIONAL ABLATIONS

In this section, we include some additional experiments to support the results in the main paper.

In Section 5 we introduced a scratchpad for mode, which orders the intermediate counts in order of frequency. This may seem overly demanding, as it requires the model to know the order of the frequencies before outputting them. Another variant of this could output the scratchpad in order of appearance in the original sequence. Moreover, we can output the token first before outputting

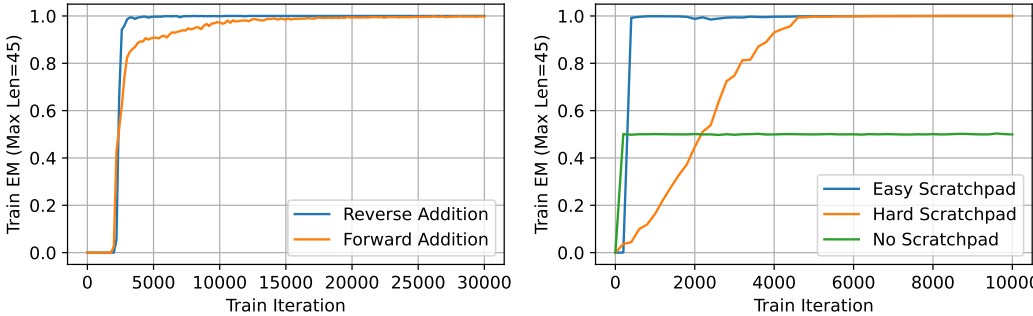

(a) Training speed comparison for addition with index hints

(b) Training speed comparison for parity with different scratchpads

Figure 6: We compare the training speed as measured by exact match on a maximum training length of 45. **(a)** compares the convergence speed of models trained on forward addition vs reverse addition with index hints. **(b)** compares the convergence speed of models trained on different scratchpads on parity.

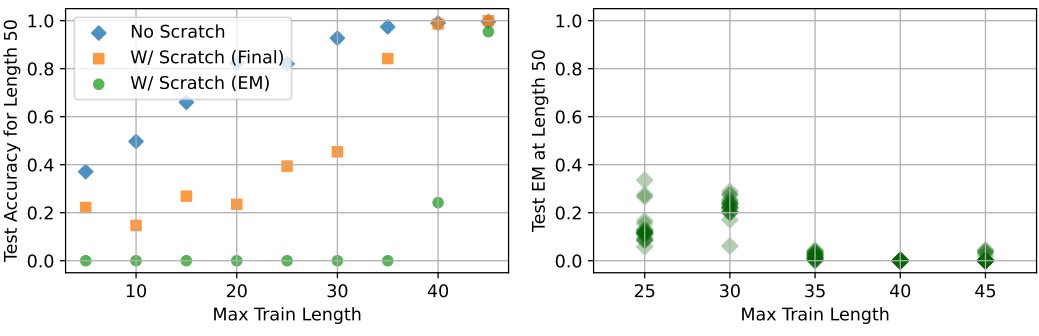

(a) Mode using scratchpad ordered by appearance

(b) Parity with sum-mod-10 scratchpad

Figure 7: **(a)** compares test performance of mode with or without scratchpad. In this case, we use the scratchpad presented in order of appearance. We see that no scratchpad significantly outperforms scratchpad, whether measured on the final answer accuracy or the exact match of the entire scratchpad output. **(b)** illustrates the generalization performance for parity with scratchpad on length 50. We see that no runs show significant length generalization in this setting.

their count, which may help the model reference the relevant token for this step. An example is

`a b b c b a c b > a 2 b 4 c 2 b` .

The performance of this scratchpad is shown in Figure 7. We see that utilizing this scratchpad still results in much worse length generalization performance than using no scratchpad.

Lastly, we showcase the spread of different training runs on the count and mode tasks in Figure 10, and on the copy tasks in Figure 11.

## F  APPENDIX: RASP DETAILS

### F.1  RASP SPECIFICATION

Here we describe how to realize RASP as a restricted subset of Python (with numpy). First, every RASP program accepts an input sequence of length $n$, for arbitrary $n \in \mathbb{N}$, and returns an output sequence of the exact same length— just like a transformer. The restrictions on Python are: All variables are either numpy arrays of length $n$ ("sequences"), or binary matrices in $\{0, 1\}^{n \times n}$ ("se-

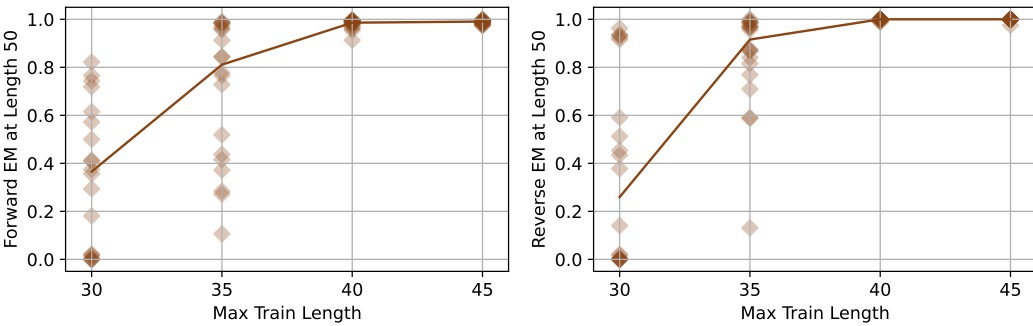

(a) Forward addition on easy carry examples

(b) Reverse addition on easy carry examples

Figure 8: Length generalization on addition with index hints. Each diamond shows the performance of one of 20 runs, illustrating the spread of different training runs. **(a)** illustrates the generalization performance for forward addition with index hints on easy carry examples of length 50. **(b)** illustrates the generalization performance for reverse addition with index hints on easy carry examples of length 50. Easy carry examples consist of addition questions where the two numbers are sampled randomly and independently, which is the setting considered in prior works studying addition. We see that both settings demonstrate strong length generalization, thus demonstrating the usefulness of the index hints.

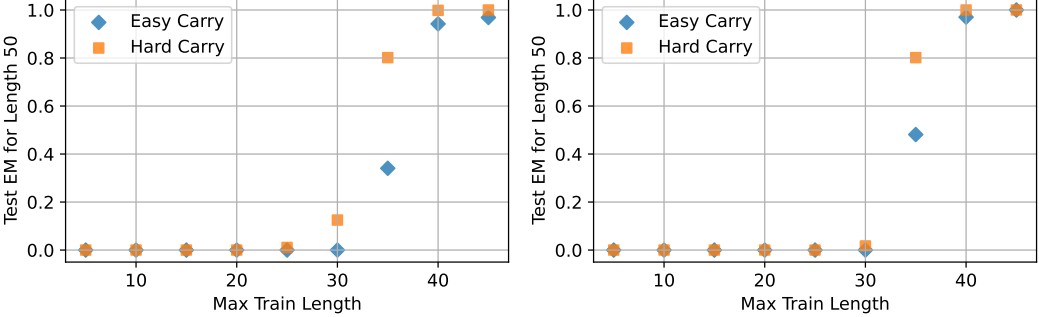

(a) Forward addition trained with balanced carry

(b) Reverse addition trained with balanced carry

Figure 9: Length generalization on addition with index hints, trained with balanced carry distribution. Each point shows the median test performance over 20 runs. **(a)** illustrates the generalization performance for forward addition with index hints on both easy and hard carry examples of length 50. **(b)** illustrates the generalization performance for reverse addition with index hints on both easy and hard carry examples of length 50. With balanced carry training, strong length generalization is observed for both forward and reverse addition on the hard carry evaluation setting. This demonstrates that with increased data diversity, we also increase the likelihood that the model learns a length-generalizing solution.

lectors"). No control flow is allowed; all programs are straight-line programs, with no branching or loops. Finally, every line of the program must be a call to either one of the core functions defined in Listing 1, or to another RASP program.

It is easy to confirm that this is equivalent to the original presentation of RASP in Weiss et al. (2021).

## F.2 RASP-L: LEARNABLE RASP

The original RASP technically allows for certain operations which are possible to represent, but not "easy" to represent or learn. For example, any arbitrarily-complex tokenwise operation $\mathbb{R} \to \mathbb{R}$

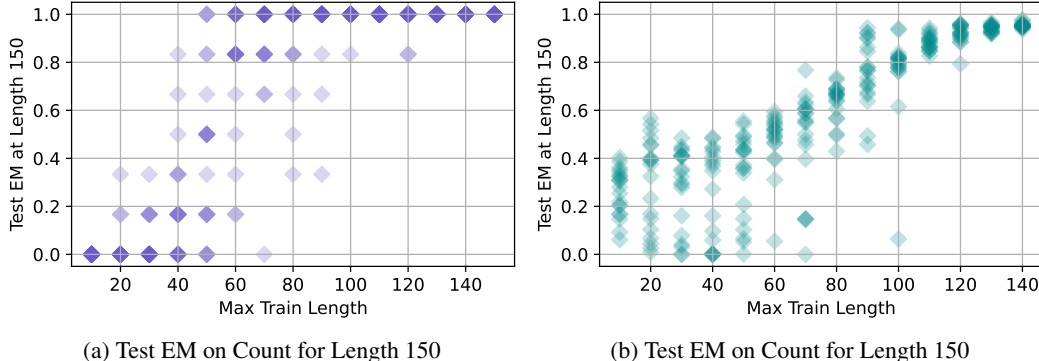

(a) Test EM on Count for Length 150

(b) Test EM on Count for Length 150

Figure 10: **(a)** illustrates the generalization performance for count with varying training length on test examples of length 150. **(b)** illustrates the generalization performance for mode with varying training length on test examples of length 150.

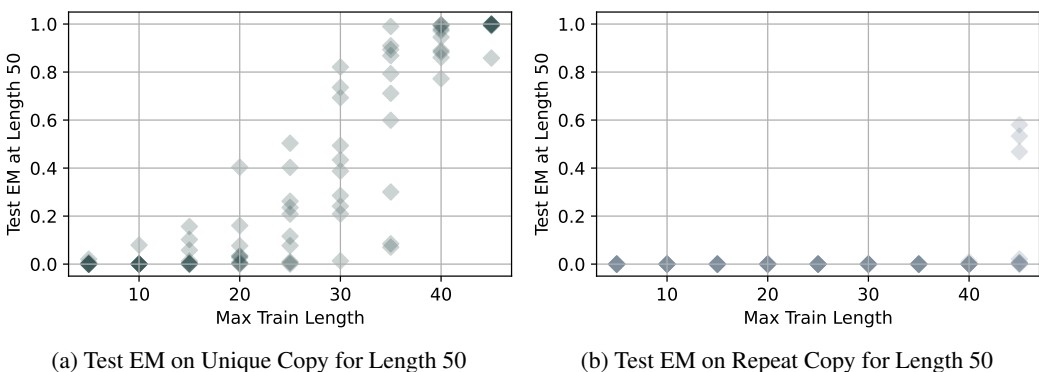

(a) Test EM on Unique Copy for Length 50

(b) Test EM on Repeat Copy for Length 50

Figure 11: **(a)** illustrates the generalization performance for unique copy with varying training length on test examples of length 50. **(b)** illustrates the generalization performance for repeat copy with varying training length on test examples of length 50.

is allowed. To disallow such pathologies, we define a "learnable" subset of RASP which we call RASP-L. RASP-L enforces the following additional restrictions on RASP.

First, all non-index values, including the input and intermediate variables, must be of type `int8`. This constraint handles issues with both infinite-precision and with learnability of the token-wise operations (since all tokenwise functions now have small finite domains and co-domains, `int8` → `int8`, and so can be easily memorized). Although disallowing floating-point operations and unbounded numbers may seem like a strong restriction, we find these are not necessary for most symbolic programs.

Moreover, token indices are treated specially. Standard RASP allows arbitrary arithmetic operations on indices, such as division-by-two: $i \mapsto \lfloor i/2 \rfloor$. However, transformers must decode index information from their positional embeddings, so any RASP-L operation on indices must be "easy" to perform on the appropriate positional embeddings. We find that empirically, such operations involving index-arithmetic are often not learned in a length-generalizing manner. This is potentially because it is difficult to learn globally valid arithmetic structures (such as division by two) from essentially local examples, i.e. short contexts. To reflect this in RASP-L, we only allow the following operations on indices: order comparisons with other indices, and computing successor/predecessor. Formally, the RASP-L core function `indices(x)` returns a special type `IndexInt`, which can take values in $\mathbb{N}$, but only allows these restricted operations. That is, we allow adding 1 to an

```python
def map(x, func):
    return np.array([func(xi) for xi in x])

def seq_map(x , y, func):
    return np.array([func(xi, yi) for xi, yi in zip(x,y)])

def select(k, q, pred):
    s = len(k)
    A = np.zeros((s, s), dtype=bool)
    for i in range(s):
        # k_index <= q_index due to causality
        for j in range(i+1):
            A[i, j] = pred(k[j], q[i])
    return A

def sel_width(A):
    return np.dot(A, np.ones(len(A)))

def aggr(A, v):
    out = np.dot(A, v)
    norm = sel_width(A)
    return np.divide(out, norm, out=np.zeros_like(v), where=(norm != 0))

def indices(x):
    return np.arange(len(x))

def fill(x, const):
    return np.array([const] * len(x))

# Convenience function:
def kqv(k, q, v, pred):
    return aggr(select(k, q, pred), v)
```

Listing 1: RASP-L core functions.

IndexInt, but we do not allow adding two IndexInts, nor casting between IndexInt and int8.

There is one additional restriction on RASP, involving the "selector width" core operation. Selector-width in standard RASP returns the number of prior elements that are selected by a binary Attention matrix, for each token position. The return type of Selector-width in RASP-L inherits from IndexInt: thus it can represent unbounded numbers of selected elements, but can only operate on them in restricted ways. Moreover, every call to sel_width in RASP-L returns a *new type* which inherits from IndexInt, and these types cannot be compared to each other. That is, the sequences returned by two different calls to sel_width are incomparable. The reason for these restrictions, which may otherwise seem contrived, is that sel_width can be used to simulate indices, by calling it on the all-ones selector matrix. Thus, we must restrict the output of sel_width sufficiently to not allow bypassing the intended restrictions on index-arithmetic. There may also be more mechanistic motivations for such restrictions, since the Transformer implementation of selector width requires weights which grow linearly with sequence length (Lindner et al., 2023).

## F.3 RASP Programs

In this section, we provide the RASP-L core (1) and library (2); RASP-L programs for the tasks discussed in the paper, namely counting (3), mode (4), copy-with-unique-tokens (5), addition with reverse-order and index-hints (7); and a naive non-RASP addition algorithm (6).

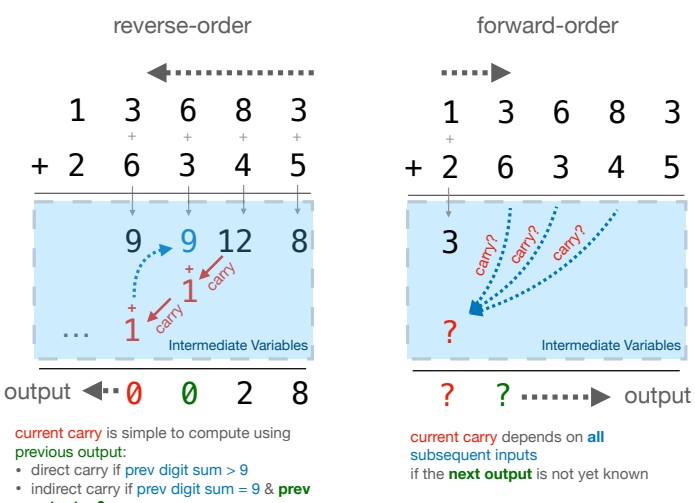

Figure 12: Intuition for why reverse order addition is simpler to implement in RASP than forward order. In reverse order, the carry calculation is simple with the help of the most recent output digit, which acts like a built-in scratchpad. In forward order, the carry requires a more complex calculation involving all remaining input digits. This intuition is reflected in the RASP-L program for forward addition, which is much longer than that of reverse addition (see Listings 7 and 8).

```python
def where(condition, x_if, y_else):
    # equivalent to np.where(condition, x_if, y_else)
    x_masked = seq_map(x_if,    condition, lambda x, m: x if m else 0)
    y_masked = seq_map(y_else,  condition, lambda y, m: y if not m else 0)
    return seq_map(x_masked, y_masked, lambda x, y: x if y == 0 else y)

def mask(x, bool_mask, mask_val=0):
    # equivalent to x*bool_mask + default*(~bool_mask)
    return where(bool_mask, x, fill(x, mask_val))

def cumsum(bool_array):
    # returns number of previous True elements in bool_array
    return sel_width(select(bool_array, bool_array, lambda k, q: k))

def shift_right(x, n, default=0):
    # shifts sequence x to the right by n positions
    return kqv(indices(x) + n, indices(x), x, equals, default=default)

def maximum_xv(x, v):
    # returns out[i] = v[argmax(x_{<= i})]
    which_gt = select(x, x, gt)
    num_gt = sel_width(which_gt)
    highest = (num_gt == 0)
    idx_hold = cumsum(highest)   # zero-order hold of the number of the highest incides so far.
    idx_hold_if_highest = seq_map(idx_hold, highest, lambda i, h: i if h else 0)
    z = kqv(idx_hold_if_highest, idx_hold, v, equals)
    return z

def maximum(x):
    return maximum_xv(x, x)

def minimum(x):
    return -maximum(-x)

def argmax(x):
    return maximum_xv(x, indices(x))

def argmin(x):
    return argmax(-x)

def num_prev(x, queries):
    # output[i] = number of previous elements of x equal to queries[i], inclusive
    return sel_width(select(x, queries, equals))

def firsts(x, queries, default=0):
    # find the index of the first occurrence of each query[i] in x
    # out[i] := np.flatnonzero(x[:i+1] == queries[i]).min()
    NULL_VAL = -1 # special token, cannot appear in x
    has_prev = kqv(shift_right(x, 1), x, fill(x, 1), equals) # if x[i] has occured previously
    first_occ = 1-has_prev
    first_occ_only = where(first_occ, x, fill(x, NULL_VAL)) # out[i] = x[i] if first_occ[i] else NULL_VAL
    return kqv(first_occ_only, queries, indices(x), equals, default=default)

def find_last_tok(x, tok):
    # finds the index of the last occurrence of tok in sequence x (causally, inclusive)
    toks = fill(x, tok)
    matches = (x == toks)
    nprev = sel_width(select(x, toks, equals))
    nprev_matches_only = mask(nprev, matches) # zero-out non-toks
    idxs = kqv(nprev_matches_only, nprev, indices(x), equals)
    return idxs

def index_select(x, idx, default=0):
    # indexes into sequence x, via index sequence idx
    # i.e. return x[idx] if idx[i] <= i else default
    return kqv(indices(x), idx, x, equals, default=default)

def first_true(x, default=-1):
    # returns the index of the first true value in x
    seen_true = kqv(x, fill(x, 1), fill(x, 1), equals, default=0)
    first_occ = seq_map(seen_true, shift_right(seen_true, 1), lambda curr, prev : curr and not prev)
    return kqv(first_occ, fill(x, 1), indices(x), equals, default=default)

def induct(k, q, offset, default=0, null_val=-999):
    # get value of k at index of first occurrence of q (if found) plus offset
    # null_val is a special token that cannot appear in k or q; used to prevent accidental matches
    indices_to_copy = firsts(shift_right(k, offset, default=null_val), q, default=null_val)
    # copy values of k at indices_to_copy (use requested default for invalid indices)
    copied_values = index_select(k, indices_to_copy, default=default)
    return copied_values
```

Listing 2: RASP-L library functions.

```python
def count(seq):
    # First, find the index of most-recent START_TOK (beginnng of current sequence)
    start_idx = find_last_tok(seq, START_TOK)

    # Then, compute the start/end numbers of the current sequence
    start_nums = index_select(seq, start_idx+1)
    end_nums   = index_select(seq, start_idx+2)

    # Bool arrays: whether we're predicting the first / last tokens of the current sequence
    pred_first_pos = seq_map(indices(seq), start_idx+2, equals)
    pred_final_pos = (~pred_first_pos) & seq_map(seq, end_nums, equals)

    next_tok = where(pred_first_pos,            # if predicting the first token:
                     start_nums,                #    next_tok = starting num
                     where(pred_final_pos,      # else if predicting the final token:
                           full(seq, END_TOK),  #    next_tok = END_TOK
                           seq + 1))            # else: next_tok = prev_tok + 1
    return next_tok
```

Listing 3: RASP-L program for Count.

```python
def mode(x):
    num_prev_matching = sel_width(select(x, x, equals))
    idx = argmax(num_prev_matching)
    return index_select(x, idx)

def binary_mode(x):
    num_prev_zeros = sel_width(select(x, fill(x, 0), equals))
    num_prev_ones  = sel_width(select(x, fill(x, 1), equals))
    mode_val = seq_map(num_prev_ones, num_prev_zeros, gt)*1
    return mode_val
```

Listing 4: RASP-L program for Mode.

```python
def copy_unique(seq):
    return induct(seq, seq, offset=1)

def copy_unique_ar(x):
    START, END = -1, -2
    prompt = np.concatenate(([START], x, [END], [START]))
    seq = prompt.copy()

    while seq[-1] != END:
        next_tok = copy_unique(seq)[-1]
        seq = np.concatenate((seq, [next_tok]))
    return seq

copy_unique_ar(np.array([8, 3, 4, 2, 1, 5]))
>> [-1  8  3  4  2  1  5 -2 -1  8  3  4  2  1  5 -2]
```

Listing 5: RASP-L program for Copy with unique tokens.

```python
def add_illegal(inp):  # inp = array of zero-padded digits of x0, x1; returns z = x0 + x1
    num_digits = int(len(inp)/2)  # ILLEGAL: no division on index types
    z = np.zeros(num_digits)
    carry = 0
    reversed_range = range(num_digits)[::-1]  # ILLEGAL: reversal is non-causal
    for i in reversed_range:  # ILLEGAL: no for loops
        x0, x1 = inp[i], inp[num_digits+i]  # ILLEGAL: variables cannot be used as indices
        digit_sum = x0 + x1
        z[i] = (digit_sum + carry) % 10
        carry = 1 if digit_sum > 9 else 0
    return z
```

Listing 6: An addition program that is illegal in RASP-L for several reasons.

```
1    ## Constants and helper functions
2    START_PROMPT = -1
3    PLUS = -2
4    EQUALS_SIGN = -3
5    END_RESPONSE = -5
6    NONE = -88
7
8    def mask_between_tokens(seq, tok0, tok1):
9        seen_tok0 = has_seen(seq, full(seq, tok0))
10       seen_tok1 = has_seen(seq, full(seq, tok1))
11       ind_between = seq_map(seen_tok0, seen_tok1, lambda a, b: a and not b)  # ind(tok0) <= (*) < ind(tok1)
12       return ind_between
13
14   def _add_safe(x, y):
15       return x + y if (x >= 0) else x # preserve index-hints
16
17   ## Next-token function
18   def next_tok_rev_addition_hinted(seq):
19       prompt_mask = 1-has_seen(seq, full(seq, EQUALS_SIGN))
20       second_summand_mask = mask_between_tokens(seq, PLUS, EQUALS_SIGN)
21       prompt = mask(seq, prompt_mask)
22
23       # let's first align the 1st summand with the second.
24       other_summand_digit = induct(k=prompt, q=shift_right(prompt, 1), offset=1)
25       pairsums = seq_map(seq, other_summand_digit, _add_safe)  # this aligns pairsums with the 2nd summand
26       pairsums = mask(pairsums, second_summand_mask, NONE)
27       pairsums_nh = mask(pairsums, (seq >= 0), NONE) # no hints: only keep digits
28
29       curr_output_digit  = shift_right(seq, 1)
30       curr_pairsum = induct(pairsums, shift_right(seq, 2), offset=1) # pairsum that generated curr_output_digit
31       next_pairsum = induct(pairsums, seq, offset=1)
32
33       ## START CHANGES
34       direct_carry = curr_pairsum > 9  # previous sum gives carry
35       indirect_carry = (curr_pairsum == 9) & (curr_output_digit == 0)  # prev sum is 9, earlier sum gave carry
36       next_tok_gets_carry = direct_carry | indirect_carry
37
38       # (simple) index-hint computations:
39       final_hint = full(seq, -100) # final hint output is always -100
40       first_hint =  induct_prev(seq, full(seq, EQUALS_SIGN), offset=-2) # first hint is 2 places before '='
41       next_hint = shift_right(seq, 1) + 1
42       eos = (next_hint > final_hint)
43       ## END CHANGES
44
45       next_tok = next_pairsum
46       next_tok += next_tok_gets_carry
47       next_tok = next_tok % 10
48
49       ## Finally, handle the case of outputing index-hints
50       next_tok_is_index_hint = (seq > -100) # all index-hints are <= -100
51       eos = (eos & next_tok_is_index_hint)
52
53       next_tok = where( next_tok_is_index_hint, next_hint, next_tok)
54       next_tok = where( eos, full(seq, END_RESPONSE), next_tok)
55       next_tok = where( (seq == EQUALS_SIGN), first_hint, next_tok)
56       return next_tok
```

Listing 7: RASP-L program for addition, with output in reverse order, and index-hints. See Section 5.1 for details on prompt format. For addition in forward order, the highlighted codeblock is replaced with Listing 8.

```
1    ## START CHANGES
2    gives_carry = tok_map(pairsums_nh, lambda _x: 1 if _x > 9 else 0)
3    z = cumsum((pairsums_nh != 9) & (pairsums_nh != NONE))
4    u = mask(z, gives_carry, mask_val=NONE)
5    v = tok_map(u, lambda _x: _x - 1)
6    chain_end_idxs = firsts(z, v, default=NONE)   # (left) ending indices of carry-chain
7
8    curr_tok_got_carry = ((curr_pairsum % 10) != curr_output_digit)
9    next_tok_inside_carry_chain =  (next_pairsum == 9) & curr_tok_got_carry
10       # in the middle of a carry-chain? (NOTE: assumes the pairsums has first element 0)
11
12   next_tok_idx = kqv(pairsums, seq, indices(seq), equals) + 1
13       # which answer-position are we at? (indices aligned to pairsums)
14   next_tok_chain_end = kqv( chain_end_idxs , next_tok_idx , full(seq, 1), equals, default=0)
15       # does the next_tok get a carry from the end of a carry-chain?
16   next_tok_gets_carry = next_tok_inside_carry_chain | next_tok_chain_end
17
18   # (simple) index-hint computations:
19   final_hint = induct_prev(seq, full(seq, EQUALS_SIGN), offset=-2) # final hint is 2 places before '='
20   first_hint = full(seq, -100)
21   next_hint = shift_right(seq, 1) - 1
22   eos = (next_hint < final_hint)
23   ## END CHANGES
```

Listing 8: The required patch to Listing 7, to produce a program for addition in forward order. This code replaces the highlighted block in Listing 7. See Section 5.1 for details on prompt format.

## G  LIMITATIONS AND OPEN QUESTIONS

**Limitations of RASP-L.**   The RASP and RASP-L programming languages were remarkably useful in our work, for reasoning about which algorithms are easy to represent and learn. However, they have clear limitations. For one, RASP and RASP-L are not *complete*: not all functions which can be efficiently represented by Transformers have efficient RASP implementations. A notable class of Transformer algorithms which are ill-suited for RASP are numerical algorithms, such as in-context gradient-descent and linear regression, which involve high-dimensional matrix and vector floating-point operations (Akyürek et al., 2022; Garg et al., 2022; Charton, 2021). Moreover, RASP supports only deterministic outputs and binary attention masks. In addition, some of the RASP-L restrictions may seem somewhat contrived (such as no floating-points and index restrictions)— there may be a more natural way to capture the set of easily-representable algorithms for standard Transformers. Nonetheless, we view RASP and RASP-L as important steps towards reasoning about Transformer algorithms, and we hope to see future work in this direction.

**Limitations of Complexity Measures.**   We considered for simplicity a basic notion of function complexity, which is the minimum RASP-L program length. This intuitive notion captures many empirical behaviors, as we demonstrated. However, depending on the operations, it can be possible to represent multiple lines of RASP-L in a single layer of a Transformer, and so RASP program length does not perfectly correspond to Transformer-complexity. There are likely more refined notions of complexity, such as the "parallel depth" of RASP-L programs, or lower-level measures like the minimum-weight-norm among all weight-noise-robust Transformers which implement the function. Many of these notions, including ours, have the drawback of being likely intractable to compute— intuitivly, it may be difficult to find the minimum RASP-L program for a task for similar reasons that Kolmogorov complexity is uncomputable (Kolmogorov, 1963). We leave investigating more refined notions of complexity to future works.

**On Formal Language Characterizations.**   One important question that remains open is whether there exists a natural complexity-theoretic definition of the class of tasks which are "simple" for autoregressive Transformers to represent. For example, it is well-known that RNNs tend to length generalize on tasks equivalent to *regular languages*, like Parity (e.g. Delétang et al. (2023)). This is intuitively because regular languages can be decided by a class of algorithms (deterministic finite automata) which are "simple" to represent by RNNs, and thus plausibly easy to learn. We would like an analogous characterization of which tasks admit algorithms with a simple and natural Transformer representation. Recent works have characterized which functions are *possible* to represent by a Transformer[6], but this representation is not always "simple" enough to be learnable, and not always uniform (Hahn, 2020; Merrill et al., 2022; Chiang & Cholak, 2022; Pérez et al., 2021; Bhattamishra et al., 2020; Ebrahimi et al., 2020; Merrill & Sabharwal, 2023; Sanford et al., 2023). Our presentation of RASP-L is meant to be one way of defining algorithms which are "simple" to represent— those expressable as short RASP-L programs— but this definition is not explicit (and likely not complete).

## H  COMPARISON TO MIN-DEGREE-INTERPOLATORS

An essential aspect of our work is our Transformer-specific notion of function complexity: the minimum RASP-L program length. Here we show why this choice is important, by contrasting it with another popular notion of complexity: minimum polynomial degree. Concretely, Abbe et al. (2023) recently proposed a model of learning in which Transformers learn the minimum-degree function which interpolates their train set. We will give a simple example where our RASP toy model correctly predicts a Transformer's out-of-distribution generalization behavior, but the min-degree-interpolator model does not. We emphasize that these results are not inconsistent with Abbe et al. (2023): neither Abbe et al. (2023) nor our current work claim to apply in all settings. Rather, this section illustrates how a Transformer-specific measure of complexity can be more predictive than architecture-agnostic measures of complexity, in certain settings.

---

[6]Note that the exact set of which functions are representable depends on certain definitional details of Transformers such as finite vs. infinite precision, bounded vs. unbounded weights, etc, which is why some of these references arrive at different conclusions.

## H.1 THE SETTING: BOOLEAN CONJUNCTION

We consider the "Generalization-on-the-Unseen" setting of Abbe et al. (2023), for direct comparison. Our target function is simply boolean conjunction. Given $n = 20$ input bits $x_i \in \{0, 1\}$, the ground truth function is the boolean AND of all bits: $f^*(x_1, x_2, \ldots, x_n) = \bigwedge_{i \in [n]} x_i$. That is, $f^*(x) = 1$ iff all bits $x_i = 1$, and $f^*(x) = 0$ otherwise. Now, the train distribution is supported on inputs $x$ where the last $k = 5$ bits of $x$ are always 1. Specifically: with probability $1/2$, $x$ is the all-ones vector, otherwise $x$ is all ones with a single 0 in a random location among the first 15 bits. That is,

$$x \sim (1^n - B(1/2)e_i); \quad i \sim \text{Unif}[1, 15].$$

where $e_i \in \{0, 1\}^n$ is the $i$th standard basis vector. Note the ground-truth label $y = f^*(x)$ for this distribution is balanced. The unseen test distribution is identical to the train distribution, except the '0' is only among the last 5 bits. That is,

$$x^{(\text{Unseen})} \sim (1^n - B(1/2)e_i); \quad i \sim \text{Unif}[16, 20].$$

We now ask: when a Transformer is trained on the above train distribution, what does it predict on the unseen test distribution?

**Experimental Result.** We train a standard decoder-only Transformer autoregressively in the above setting, with sequence distribution $[x, f^*(x)]$ for $x$ sampled from the train distribution. (Experimental details are given below.) We find that the trained Transformer reaches 100% test accuracy on the unseen test set. That is, the Transformer correctly computes the boolean AND, even on bits which were irrelevant at train time.

Experimental detials: For this task, we train on sequences of length 20 and evaluate on test sequences of length 20. We train a 2 layer, 4 head autoregressive Transformer, with embedding dimension 64. We train with Adam for 10000 iterations, using a context size of 128, batch size 128, and learning rate 1e-3. To compare to the setting in Abbe et al. (2023), we do not pack the context here and train on single examples in the context window. The training distribution has a 50% chance of being a sequence of all 1s, and a 50% chance of having one 0 element in the sequence. The position of the 0 element is sampled uniformly between positions 0 and 15. The last 5 elements in the training sequence are always 1s. At test time, there is a 50% chance of being a sequence of all 1s, and a 50% chance of having one 0 element in the last 5 elements in the sequence.

## H.2 THE MIN-DEGREE INTERPOLATOR

We now claim that the minimum-degree-interpolator of the train set *does not* behave like the Transformer in the above experiment. To see this, observe that the minimum-degree-interpolator will not depend on the last $k = 5$ bits of the input, since these bits are constant on the train set. This can be formalized via the following simple lemma, which uses the same notion of "degree profile" (DegP) as Abbe et al. (2023)[7].

**Lemma 1.** *For all subsets $S \subseteq \{0, 1\}^n$ and all boolean functions $f : \{0, 1\}^n \to \mathbb{R}$, the following holds. Let $g^* : \{0, 1\}^n \to \mathbb{R}$ be the boolean function of minimum degree-profile which agrees with $f$ on $S$. That is,*

$$g^* = \underset{\substack{g:\{0,1\}^n \to \mathbb{R} \\ s.t. \ g|_S = f|_S}}{\arg\min} \ \text{DegP}(g).$$

*Let $I \subseteq [n]$ be the subset of indices (if any) on which $S$ is constant. That is, $\pi_I(S)$, the projection of $S$ to coordinates $I$, is a singleton. Then, the minimum-degree-interpolator $g^*$ also does not depend on indices $I$. That is,*

$$x_i = y_i \forall i \notin I \implies g^*(x) = g^*(y).$$

This lemma follows from the fact that if an interpolator depends on bits which are always constant on the train set $S$, then its degree-profile could be reduced without changing its output on $S$, and thus it cannot be a min-degree-interpolator. For completeness, we state this formally below.

---

[7]Briefly, the degree profile of a boolean function is the tuple of its Fourier weights at each level, with the natural total ordering which refines the standard polynomial degree.

**RASP-Length.** On the other hand, there is a one-line RASP-L program which computes the boolean AND of all input bits:

```
def output(x): kqv(x, full(x, 0), full(x, 0), equals, default=1)
```

This program can be represented by a 1-layer Transformer, where the attention layer simply "searches for a 0" in the prior context. Thus, our RASP toy model of learning predicts the correct experimental result in this setting. The intuition here is that it is actually "easier" for a Transformer to represent the boolean conjection of *all* input bits, and so it learns to do so. Learning to ignore the last 5 bits would have been possible, but is an additional complication, and does not help in fitting the train set in any case. Notably, this intuition (and indeed, the experimental result) does not hold for MLPs; it is specific to the Transformer architecture.

### H.2.1 PROOF OF LEMMA

Lemma 1 follows directly from the following fact. We state this with the boolean hypercube identified with $\{\pm 1\}^n$ as is standard in boolean function analysis, but these statements can trivially be translated to $\{0, 1\}^n$.

**Lemma 2.** *Let $f : \{\pm 1\}^n \to \mathbb{R}$ be a boolean function. Suppose $f$ depends on its first coordinate; that is suppose $\exists z \in \{\pm 1\}^{n-1} : f(1 \circ z) \neq f(-1 \circ z)$. Then, restricting the first coordinate of $f$ to be $1$ strictly reduces its degree profile:*

$$\mathrm{DegP}(f|_{x_1=1}) < \mathrm{DegP}(f)$$

*Proof.* (Sketch). Consider the multilinear representation of $f$, and factor out terms containing $x_1$:

$$f(x_1, \overline{x}) = x_1 P(\overline{x}) + Q(\overline{x}) \tag{1}$$

where $\overline{x}$ denotes $(x_2, x_3, \ldots, x_n)$. When restricted with $x_1 = 1$, we have

$$f(x_1, \overline{x})|_{x_1=1} = P(\overline{x}) + Q(\overline{x}) \tag{2}$$

Now, because $f$ depends on $x_1$ by assumption, we must have $P \neq 0$. Thus, comparing the above two, at least one monomial containing $x_1$ has reduced degree with the restriction. The conclusion follows. $\square$

