# OpenReview forum: "What Algorithms can Transformers Learn? A Study in Length Generalization"
_ICLR.cc/2024/Conference — ICLR 2024 poster_

### Official Review · Reviewer_LyN1 · 2023-10-22

**Soundness:** 3 good
**Presentation:** 4 excellent
**Contribution:** 3 good
**Rating:** 8
**Confidence:** 3

**Summary:**

This paper focuses on the standard decoder-only transformers model for algorithmic tasks. Prior work has shown that standard transformers exhibit little length generalization on certain arithmetic tasks. Accordingly, the author first propose the RASP Conjecture to describe the kind of problems on which the transformers can perform length generalizations. The author also present empirical evidence to support the RASP Conjecture and length generalization. Finally, the author analyzes the hard tasks for transformers, such as parity and decimal addition, and proposes particular scratchpad formats to enhance the length generalization on these tasks.

**Post rebuttal update**

Having reviewed the author's response, I am impressed by the meticulous effort the authors have done in answering questions and refining the manuscript. The work has effectively addressed my initial concerns. As a result, I have raised my point.

**Strengths:**

* The paper is well-written: discussions and presentations are clear.

* The RASP Conjecture is well-motivated.

* The case study clearly explains why certain tasks are solvable (i.e., length generalizable) and why others aren't. For each solvable task, there is a corresponding RASP format.

* The author designs particular scratchpad formats for parity and decimal addition and experimentally verifies the effectiveness of these designs.

**Weaknesses:**

* The scratchpad designs for parity and decimal addition, though effective, are a bit different from the RASP format. For example, it seems that the proposed scratchpad for addition is motivated by correcting the failure modes discussed in section 4.3. It would be nice if the author could clarify the connection between the scratchpad designs and the RASP.

* This paper classifies the algorithmic tasks into easy and hard ones. The easy ones are solvable (i.e., length generalizable) by the standard transformers. As a general question, can the author recommend the possible directions to solve the hard ones? The proposed scratchpad solution requires a detailed understanding of the failure modes for each hard task, which doesn't seem quite scalable.

* Missing citations. Transformers for algorithmic tasks is an interesting topic in the recent literature. It would be nice if the authors could discuss a few more papers. From the modeling perspective, [1] introduces certain twists on the transformer architecture to enable the length generalization on algorithmic/regular language tasks such as Even Pairs, Modular Arithmetic, Parity Check, and Cycle Navigation (i.e., the tasks in [2]). From the task perspective, [3] evaluates the model performance over randomly generated finite-state transduction tasks. The idea of evaluating over randomly generated tasks is complementary to the usual conduct of evaluating on particular tasks.


[1] Chi, T.C., Fan, T.H., Rudnicky, A.I. and Ramadge, P.J., 2023. Transformer Working Memory Enables Regular Language Reasoning and Natural Language Length Extrapolation. arXiv preprint arXiv:2305.03796.

[2] Delétang, G., Ruoss, A., Grau-Moya, J., Genewein, T., Wenliang, L.K., Catt, E., Cundy, C., Hutter, M., Legg, S., Veness, J. and Ortega, P.A., 2022. Neural networks and the chomsky hierarchy. arXiv preprint arXiv:2207.02098.

[3] Valvoda, J., Saphra, N., Rawski, J., Williams, A. and Cotterell, R., 2022, October. Benchmarking compositionality with formal languages. In Proceedings of the 29th International Conference on Computational Linguistics (pp. 6007-6018).

**Questions:**

See the weakness section.

---

> ### Author Response · Authors · 2023-11-16
> **Author response**
>
> Thank you for your time and effort in reviewing our paper! We are delighted that you found our conjecture well-motivated and experimental results compelling. We are grateful that you support the acceptance of this work. We would like to address the comments and questions you’ve raised, and we’d be very open to further discussion.
>
> >The scratchpad designs for parity and decimal addition, though effective, are a bit different from the RASP format. For example, it seems that the proposed scratchpad for addition is motivated by correcting the failure modes discussed in section 4.3. It would be nice if the author could clarify the connection between the scratchpad designs and the RASP.
>
> Thank you for pointing this out. We have restructured Section 5 to better explain the connection with RASP conjecture. To summarize, one consequence of the RASP conjecture is that by converting the task into one that has a simple RASP-L solution, we can improve length generalization performance. We can do so by changing the task format, adding a scratchpad, or both. For addition, the standard format does not have a RASP-L solution because it requires index arithmetic, which is forbidden in RASP-L (see Appendix F.2). However, if we add index hints, we can then replace the operations that require index arithmetic (forbidden) with ones that use induction heads (easy to write in RASP-L) to perform indexing. Thus, the new task format can now admit RASP-L solutions. Similarly, this gives us a perspective on why scratchpad has often been found helpful for reasoning problems. In the case of parity, we cannot represent the parity algorithm because RASP-L does not allow arbitrary loops for constant depth Transformers. However, the autoregressive nature of Transformers gives us a way to simulate a simple loop by using a scratchpad. With the scratchpad format, there now exists a RASP-L program that produces the next-scratchpad-token, without requiring any loops. We hope that this clarifies the connection between the proposed designs and the RASP conjecture.
>
>
> >As a general question, can the author recommend the possible directions to solve the hard ones? The proposed scratchpad solution requires a detailed understanding of the failure modes for each hard task, which doesn't seem quite scalable.
>
> This is indeed a challenging aspect of the framework. Currently predicting or improving length generalization entails finding a RASP-L program that solves the task. We provided a few examples that can be generally more applicable, such as adding index hints for tasks that require complex indexing operations, and scratchpad for unrolling loops in the algorithm. Nonetheless, one promising future direction is to identify “unnatural operations” as suggested by RASP-L, but which are common subroutines for a large class of algorithms, and increase the training data coverage for those operations. This may help the model to perform these operations through memorization, and leverage this to learn other algorithms in a more generalizing way.
>
> >Missing citations
>
> Thank you for the suggestions, we’ve added a more thorough discussion in Appendix A.
>
> **Conclusion:**
>
> Thank you for taking the time to read our rebuttal. If our response addresses your concerns, we kindly ask that you consider raising your score. Otherwise, please let us know about your remaining concerns/questions so we can make further improvements.

---

### Official Review · Reviewer_q9T5 · 2023-10-29

**Soundness:** 2 fair
**Presentation:** 4 excellent
**Contribution:** 3 good
**Rating:** 6
**Confidence:** 3

**Summary:**

This paper discusses an important problem: in which algorithmic task transformers are more likely to length generalize in which format.

They propose a conjecture: transformers are likely to length generalize for a task if there exists a "simple" "RASP-L" program to solve that task. (The conjecture includes other conditions to ensure almost perfect in-distribution performances.) (The "RASP" programs are easier to represent and learn by transformers in the sense that their program parameterizations encourage parallel operators and discourage sequential, for-loop, and branch operations. The "RASP-L" programs further rule out numerically unstable operations such as they forbid float values and complex token indices processing.)

The authors also provide some experimental results that empirically align with the conjecture.

**Strengths:**

This paper discusses an important topic: the length generalization of transformers. It is easy to understand with many illustrative visualizations. They test the conjecture in six simple tasks whose positive/negative results all align with the conjecture. They also show improved/degraded length generalization performances as guided by the conjecture after reformulating some tasks.

Regarding the concrete conjecture, at least some parts of it align with the reviewer's intuition and experiences. For example,
* If human beings can implement a simple RASP-L program to solve a task, we do expect better length generalization of learned transformers.
* Transformers do suffer from numerical issues in practice, either for counting tokens or modeling complex indexing mechanisms. The constraints on "learnable" in addition to "RASP" are interesting.

The reviewer also appreciates the simple trick on position embeddings for better length generalization: they concatenate all samples in a long sequence with random shifting while training transformers. This is different from the common choice, which only puts one sample in one sequence, for synthetic tasks. They find that "packing and shifting the context allows all positional embeddings to be trained, and encourages the transformer to treat all positions symmetrically," therefore, towards better length generalization of transformers.

**Weaknesses:**

The main weakness is lacking the support and validation of the conjecture. The definition of RASP-L programs is also vague and unclear.

Regarding the support and validation of the conjecture:
* There is no proof or reasoning for the conjecture. There is little evidence saying that transformers actually learned or have any relationships with the "RASP-L" program in general. According to the experimental results where "transformers mostly generalize to length at most 50 when trained with length<=40", it is also hard to believe that the true "RASP-L" programs have been learned.
* The experimental results and comparison may not be solid with confusing metrics:
   - The metrics should characterize the differences between OOD and IID, instead of the absolute performances on OOD samples. As shown in Figure 1(a), generalization is about the difference in performances between in-distribution and out-of-the-distribution samples. This is important for, e.g., the addition task, because addition with 40 digits may already be out of the representation power of a 6-layer transformer in the naive forward-order. The IID performances could be as bad as the OOD ones.
  - There is also a lack of constraints on achieving almost perfect in-distribution performances before evaluating their length generalization. "Almost perfect IID performance" is assumed in the conjecture. It is also strange to evaluate the OOD performances of random runs with imperfect IID performances. If this constraint is not enforced, the current analysis may confuse the optimization difficulties with the OOD generalization performances.
   - Similarly, the metric as "median among 20 runs" is not enough to fully characterize the transformers' performance, given their non-convex optimization nature. Other metrics, such as "maximum" or "ratio of almost perfect," could be more informative and practical.
  - The tasks are simple: from count to addition/parity.
* Ablation studies on "without the index hints" are needed.

Regarding the format of the conjecture as "RASP-L" programs,
* The learnable constraints are interesting, but there are no reasoning/proof/experiments to support their choices, such as which operators are "learnable" in which sense, how to find and choose them, and if the current set of operators can represent all "learnable" operations.
* Similarly, there is no characterization of the function space represented by the "RASP-L" programs.
* The optimization efficiency conjecture in the appendix is also poorly supported. I would suggest the authors remove it if there are no supporting results from more tasks.

**Questions:**

* How did you choose the "learnable" operators? Will the representation power of "RASP" programs be affected by this additional constraint? Can all "generalizable" tasks' solutions be represented by the "RASP-L" programs?
* Are there results on the difference between the IID and OOD performances for most of the tasks?
* Are there constraints as almost perfect IID performances before evaluating their length generalization performance? Are there results with metrics such as maximum among all runs?
* Did you achieve almost perfect IID performances for the addition task with 40 digits with the naive forward order?
* Can you explain why the length generalization performance of transformers is still limited (e.g., from length<=40 to length=50) even when the conjecture conditions are fulfilled?

---

> ### Author Response · Authors · 2023-11-15
> **Author response (part 1)**
>
> Thank you for your time and effort in reviewing our paper! We are delighted by your appreciation for the importance of this line of work and for our generalization improvements on addition and parity. We would like to address the comments and questions you’ve raised, and we’d be very open to further discussion.
>
> >The definition of RASP-L programs is also vague and unclear.
>
> We want to highlight our formal definition of RASP-L programs in Appendix F (Section F.1 defines RASP, and then Section F.2 defines RASP-L based on RASP). There we specify the allowed types and the allowed program syntax for the RASP-L language. We include an informal presentation of RASP-L in the main body for brevity and accessibility, but appreciate that this may require more clarification. We would welcome any specific pointers regarding unclear aspects of the definition as we work to improve the presentation.
>
> **Regarding OOD vs IID performance:**
>
> In all of our experiments except for parity, we only evaluate models that achieve ~100% accuracy on in-distribution test samples. This includes addition on all training lengths. Parity without scratchpad is a special case as it is well known that Transformers cannot even fit the training set for long parity sequences (e.g. Chiang & Cholak and Bhattamishra et al). All parity with scratchpad experiments are trained to convergence. We hope this fully addresses your concern.
>
> >There is no proof or reasoning for the conjecture. There is little evidence saying that transformers actually learned or have any relationships with the "RASP-L" program in general. According to the experimental results where "transformers mostly generalize to length at most 50 when trained with length<=40", it is also hard to believe that the true "RASP-L" programs have been learned.
> >Can you explain why the length generalization performance of transformers is still limited (e.g., from length<=40 to length=50) even when the conjecture conditions are fulfilled?
>
> We agree that our work has no theoretical justification for the conjecture; it is currently a purely empirical claim. Nonetheless, we view such empirical work as an important first step towards understanding new phenomena -- a step upon which theory can be built. We hope that our conjectures can eventually be formalized and proven, but we leave this (significant) endeavor for future work.
>
> (As an aside, the reviewer may be interested in our new Appendix H, where we added a theoretical example from boolean function analysis that illustrates our conjecture in a simple case).
>
> We wish to clarify the scope of our main claims. Our Main Conjecture (Section 2) only characterizes which tasks Transformers are likely to length-generalize on, and not why or how they do so. Specifically, our Main Conjecture only claims that RASP-L is a useful predictive tool: empirically, if a task can be solved by a simple RASP-L program, then it is likely to exhibit strong length generalization, and vice versa. This is a phenomenological claim, as opposed to a mechanistic one. Nonetheless, we believe that the toy model in our introduction suggests a plausible mechanism for Transformer learning, and could guide new lines of mechanistic investigations in future work.
>
> **Re. degradation at longer lengths:**
>
> There are a number of factors that could influence how robustly the correct solution is learned, if indeed it is. We do not expect (and indeed do not observe) perfect length generalization over arbitrary lengths. Some level of degradation is likely fundamental due to issues trying to approximate a discrete program with a continuous space (e.g. noisy optimization, finite precision, etc).
>
> One other possible explanation, compatible with the RASP conjecture, is that there exists other “shortcut” solutions which do not generalize, but are simple enough to compete with the true RASP-L program during training. Since it is likely intractable to determine the minimum RASP-L program that fits a given training set, we cannot predict a priori what forms of “data diversity'' are required to rule out such ”shortcuts“, even if our conjecture holds true. Nonetheless, our study focuses on the relevant characteristics of tasks that influence generalization performance, thus even good albeit imperfect generalization lends strong signal to the usefulness of this perspective. We have added several new paragraphs in the revised pdf (in Section 2, and Appendix G), which highlights these limitations for the reader.

---

> ### Author Response · Authors · 2023-11-15
> **Author response (part 2)**
>
> >the metric as "median among 20 runs" is not enough to fully characterize the transformers' performance, given their non-convex optimization nature. Other metrics, such as "maximum" or "ratio of almost perfect," could be more informative and practical.
>
> In Fig 3, 7b, and 8, we show the individual test performance of all 20 runs (plotted as points). For brevity we default to reporting the median, but most experiments follow the same pattern as those in Fig 3. We hope these plots can give you a fuller picture of Transformer’s performance. If there is a specific additional metric you would like to see, we are happy to compute it on the 20 runs shown.
>
> >The tasks are simple: from count to addition/parity.
>
> On the positive side (length-generalizing tasks), we agree that it would be interesting to demonstrate more complex tasks which length-generalize. We focused our paper on simple tasks for exposition purposes, and to better connect with the existing literature on length generalization (where such simple tasks are standard). However, our RASP-L framework is more general, and can indeed express more complex tasks than we presented here due to space constraints (such as sorting and finite state automata). We can consider adding such tasks into the full version of this paper.
>
> On the negative side, we could have included a large number of complex tasks, because most tasks will not length-generalize and will also not have a RASP-L program. A number of more complex tasks have been tested in the literature, where they were shown to fail at length-generalization. These tasks include: dynamic programming, multiplication, and certain kinds of logic puzzles (as tested in Dziri et al. 2023). We thought about these tasks, and could not devise RASP-L programs for them (even for e.g. multiplication), suggesting that they are unlikely to have simple RASP-L solutions. In general, Transformers only seem to length-generalize on very particular types of tasks, and so we focused on identifying positive instances of length generalization, since negative instances abound.
>
> >Ablation studies on "without the index hints" are needed.
>
> We mention briefly in Section 4.3 that no generalization is observed on standard addition or parity (no index hints) under our experimental setting. We evaluate different lengths in increments of 5, and all 20 runs for all length showed a test EM of ~0% on these tasks. For brevity we do not include a plot, but we have added this detail to the updated pdf for better clarity.
>
> >The learnable constraints are interesting, but there are no reasoning/proof/experiments to support their choices, such as which operators are "learnable" in which sense, how to find and choose them, and if the current set of operators can represent all "learnable" operations.
>
> We agree that our work does not provide a theoretical justification for the definition of RASP-L. However, we do consider the experiments in the paper as evidence that our RASP-L constraints are meaningful (though not conclusive of course).
> Specifically, we gave RASP-L programs for all the “symbolic tasks” which were known to length-generalize in the literature (to the best of our knowledge) — this shows our definition is expressive enough to capture positive instances. Conversely, for tasks which do not length-generalize, we were unable to devise RASP-L programs to solve them: we view this as heuristic evidence that RASP-L is not “too expressive.” We’ve added a table in Figure 1 which summarizes all our experiments, to show the RASP-L evidence more clearly.
>
> We acknowledge that we do not have proofs that tasks (such as Parity) cannot be written in RASP-L, but we note that similar statements about computational complexity classes are notoriously difficult to prove (they amount to circuit-complexity lower-bounds). However, our claims are at least falsifiable: one simply has to give a RASP-L program that solves e.g. Parity, if such a program exists.
>
> Regarding whether the current set of operators can produce all “learnable” operations: Thank you for bringing up this point. We have added a more thorough discussion of this in Appendix G, paragraph “Limitations of RASP-L.” Briefly, we believe RASP-L captures most “symbolic” learnable programs, but it certainly does not capture all learnable programs. A large class of learnable but non-RASPable tasks are numerical algorithms, such as linear regression, which involve high-dimensional matrix and floating-point operations. One could imagine extending RASP-L to encompass numerical algorithms as well — we consider this an interesting direction for future work.
>
>
> >There is no characterization of the function space represented by the "RASP-L" programs.
>
> We agree that it is an important open question, so we have added this discussion explicitly in Appendix G, paragraph “On Formal Language Characterizations.” This is also an open question even for the original RASP language of Weiss et al. (2021).

---

> > ### Author Response · Authors · 2023-11-15
> > **Author response (part 3)**
> >
> > **Conclusion:**
> >
> > Thank you for taking the time to read our rebuttal. If our response addresses your concerns, we kindly ask that you consider raising your score. Otherwise, please let us know about your remaining concerns/questions so we can make further improvements.
> >
> > **References**
> >
> > Chiang & Cholak: Overcoming a Theoretical Limitation of Self-Attention
> >
> > Bhattamishra et al: On the Ability and Limitations of Transformers to Recognize Formal Languages
> >
> > Dziri et al: Faith and Fate: Limits of Transformers on Compositionality
> >
> > Weiss et al: Thinking like Transformers

---

> > > ### Author Response · Authors · 2023-11-20
> > > **Follow up**
> > >
> > > Dear reviewer,
> > >
> > > We wanted to follow up and gently ask if we are able to address your concerns in the author response. As the discussion period is nearing an end, we would appreciate any updates or further questions you may have. We thank you for your time in advance!

---

> ### Comment · Reviewer_q9T5 · 2023-11-20
> **Need more time to read all**
>
> I appreciate the authors' efforts in the thorough and long rebuttals. Unfortunately, I haven't been able to find time to read all the rebuttals and the appendix referred to in the rebuttal. Here, I will try to frame my current thoughts in case they are useful for the authors' further rebuttal. I will try to read all the rebuttals and the referred appendix before the deadline.
>
> For concerns regarding experiments, one wield thing for me to believe is that the authors claim to achieve "perfect" or "~100%" performances for addition with 20/40-digit numbers using a six-layer transformer. The task may even be out of the representation power of the models in the reviewer's understanding. It would be great if the authors could share codes to reproduce these results for double-checking.
> * Also, my main concerns regarding the experiments are whether or not they are strong enough to support the authors' claims without more theoretical analysis. Unfortunately, the concerns still stand for e.g., the only +10 length generalization, the tasks are still simple, and some confusing points as stated above. For the metrics, just for what it's worth, I would recommend plotting figures like Fig. 3 for all experiments, or using more metrics like percentage of "perfect generalization" or maximum to show how likely the models can successfully generalize instead of the average/median performances. In some sense, we are characterizing how many local minimums are "good" instead of their averaged performances.
>
> For the definition and formulation of RASP-L, I agree it is difficult to formalize "learnable" for neural networks, but this is an important concern from my perspective, given the claims of this paper. I have not checked Appendix G yet. Hopefully, it can help alleviate some of these concerns.
>
> Overall, I appreciate the authors' efforts in studying this important and challenging problem. On the other hand, I would also like to make sure the claims and results are precise and solid. Hopefully, my concerns could be alleviated by reading more of the rebuttals and appendix.

---

> > ### Author Response · Authors · 2023-11-21
> > **A brief summary**
> >
> > We appreciate your response and your continued engagement! We believe our earlier, more detailed rebuttal should address your main concerns. To quickly provide a bit more nuance on these points:
> >
> > We can confirm that all experimental results (except on parity without scratchpad) achieve near perfect train EM on in-distribution results. We train the models large enough for long enough to ensure this, and we do not use a fixed training set, but rather sample from the training distribution each batch. We also note that we pad the two summands with 0s such that they are the same length as each other, which further simplifies the problem. This is not surprising, even for a task like addition, as we know that a RASP-L program (i.e. a simple constant depth Transformer) exists to solve this for any length (this intuition applies to in-distribution performance even without index hints, since index information do not need to length-generalize in this setting). We’ll be happy to release code with the final paper to aid with any reproducibility efforts.
> >
> > We’ve added some discussion regarding the +10 length and the tasks. In short, +10 length, although somewhat arbitrary, indicates a qualitative difference between typical length generalization behavior reported in prior work, in which performance degrades as soon as we go out of distribution. Moreover, our tasks are in line with those in prior works that study length generalization in Transformers. We hope you’ll find our detailed discussion regarding these points in the original rebuttal reasonable.
> >
> > Per your suggestion, we have added new plots in Figure 10 and 11 that show the spread over runs.
> >
> > We highlight that what we propose is still a conjecture, which is of significant interest because it empirically explains a greater number of length generalization phenomena for Transformers than other existing frameworks. We hope this will inspire further theoretical analysis and make traction towards this important and challenging problem.

---

> > > ### Comment · Reviewer_q9T5 · 2023-11-21
> > > **A quick follow-up**
> > >
> > > I would guess the authors' responses are about the inverse-order addition task, e.g., those regarding "we know that a RASP-L program exists to solve this for any length"? I was more concerned with the naive forward-order addition, in which case the model needs to predict the most significant digit directly as the first token by implicitly reasoning about the carries transferred through all 20/40 digits of both arguments. With causal attention, shared weights and representations of the models, and the large training data space, it is at least non-trivial, from my perspective, to obtain a near-perfect performance for this task with such a small model. It would be great if the authors could share more info or codes (just for this task) to help clarify.

---

> ### Author Response · Authors · 2023-11-21
> **Initial response**
>
> Thanks for the clarification question! We actually do mean there is also a RASP-L program for _forward-order_ addition. We've expanded a discussion of this in the updated Section 5.1, which we think you will find interesting. Forward addition *can* in fact generalize when we try harder, e.g. be more clever with the data diversity of the training distribution. The RASP-L programs for reverse and forward addition can be found in Listings 7 and 8. It is unintuitive that forward addition has such a solution, since the natural algorithm is done in reverse order to do the carry dependencies. However, it turns out that we can represent a similar algorithm that computes carry chains through parallel and constant step computations. We think that this is a great illustration of the usefulness and flexibility of the RASP-L perspective. The RASP-L code is hard to parse, so we'll follow up with some more intuition to help guide you through it, but hopefully this answers your question in the meantime.
>
> We also attach a code sample for the example generation part of the addition task. We removed some unnecessary components that had to do with other ablations but still apologize for the code being more complex than it needs to be.
>
>     def generate_instance(self):
>
>         if self.hard_carry:
>             # hard carry generates questions with carry chains e.g. 9999 + 0003 = 10002
>             num1_len = np.random.randint(self.min_range, self.max_range+1, size=1)[0]
>             num2_len = num1_len
>         else:
>             num1_len = np.random.randint(self.min_range, self.max_range+1, size=1)[0]
>             num2_len = np.random.randint(self.min_range, self.max_range+1, size=1)[0]
>
>         num1 = [np.random.randint(self.min_num, 10, size=1)[0]]
>         if num1_len > 1:
>             num1 = num1 + list(np.random.randint(self.min_num, 10, size=num1_len-1))
>
>         if self.hard_carry:
>             num2 = [9-x for x in num1]
>             num2[-1] = np.random.randint(num2[-1]+1, 10, 1)[0]
>         else:
>             num2 = [np.random.randint(self.min_num, 10, size=1)[0]]
>             if num2_len > 1:
>                 num2 = num2 + list(np.random.randint(self.min_num, 10, size=num2_len-1))
>
>         num1 = ''.join(map(str, num1))
>         num2 = ''.join(map(str, num2))
>
>         ndigits = max(len(num1), len(num2)) + 1 # add one to the max (for the carry to be easier)
>         s1, s2 = list(map(lambda q: q.rjust(ndigits, '0'), (num1, num2)))  # padding
>
>         start_id = np.random.randint(0, len(list(self.tokenizer.count_id.values()))-ndigits+1)
>         ids = list(self.tokenizer.count_id.values())[start_id : start_id + ndigits] # contiguous sequence of ids
>
>
>         if self.index_hints:
>             s1_sequence = ' '.join([f'{a} {b}' for a, b in zip(s1, ids)])
>             s2_sequence = ' '.join([f'{a} {b}' for a, b in zip(s2, ids)])
>         else:
>             s1_sequence = ' '.join(list(str(s1)))
>             s2_sequence = ' '.join(list(str(s2)))
>
>         prompt_sequence = f'{s1_sequence} {self.tokenizer.ADD_TOKEN} {s2_sequence}'
>
>         carry = 0
>         sumpairs = []
>         i = 0
>         for d1, d2 in zip(s1[::-1], s2[::-1]): # for all digits (LSB first)
>             n1, n2 = int(d1), int(d2)
>             ans = (carry + n1 + n2)
>             dAns = ans % 10
>             newCarry = 1 if ans >= 10 else 0
>
>             id = ids[ndigits-i-1]
>             sumpairs.append((id, str(dAns)))
>             i += 1
>
>         if not self.reversed_ans:
>             sumpairs = sumpairs[::-1]
>
>         if not self.answer_index_hints:
>             sumstr = [c for (id, c) in sumpairs]
>         else:
>             sumstr = [c for pair in sumpairs for c in pair] # flatten id, digit
>         answer_sequence = ' '.join(sumstr)
>
>         final_sequence = f'{prompt_sequence} {self.tokenizer.SEP_TOKEN} {answer_sequence}'
>         return final_sequence

---

> > ### Author Response · Authors · 2023-11-21
> > **Intuition of RASP-L for forward addition**
> >
> > First, we add each digit-position individually ("pairsums_nh").
> >
> > There are 2 types of carries:
> >
> > (A) A "direct carry": the next position generates a carry, regardless of any other digits. ("next position" means one place value in the direction of the Least-Significant-Digit).
> >
> > (B) An "indirect carry": the next position doesn't generate a carry itself, but some future position generates a "carry chain" that ends at the current position.
> >
> > Carry Type (A) is easy to compute (just check if the next pairsum is > 9).
> >
> > Carry Type (B) is trickier. The basic idea is: A carry chain means the next pairsum must be exactly =9, and may be followed by a string of consecutive 9s. We want to know if, in the pairsums, this string of 9s terminates with a number >= 10. This is the only situation in which the carry will "propagate" to the current digit.
> > These computations are possible to do in parallel: We first construct a boolean mask for when (pairsum == 9), which is trivially parallel. Then, roughly, we can compute the endpoint a "string of 9s" by looking at boundary transitions (1—>0) in this array. We can compute these boundaries in parallel, and then read the pairsum at the end of the boundary. This is roughly the intuition for the RASP-L program (though the details differ slightly).
> >
> > Note that Addition was known to be in uniform AC0 -- that is, Addition has an constant-depth poly-width circuit (Pippenger, 1987).
> > The Pippenger construction handles carries in a similarly non-trivial way as our RASP-L program, since it cannot afford linear-depth to propogate carries.
> >
> > Nicholas Pippenger. The complexity of computations by networks.

---

> > > ### Comment · Reviewer_q9T5 · 2023-11-23
> > >
> > > Thank you for all the thorough replies, the additional results, appendix, and clarifications. I do cherish this paper more with them. I also appreciate the authors' discussions of the limitations of the conjecture.
> > >
> > > Overall, I find that many of the discussions and small findings in this paper are interesting and inspiring. I would like to acknowledge them. Even though I am still worried that the main conjecture and the overall framing are that well supported, with the additional limitation discussion in Section 2 and Appendix G, I will increase my rating to 6: marginally above the acceptance threshold.
> > >
> > > I would suggest the authors discuss more of the limitations of the conjecture in the main body, as discussed in Appendix G (e.g., limited expressiveness and insufficient complexity measure). They will not diminish the contributions but will give the readers a better understanding of the problem. I also hope the authors will release the codes as stated in the rebuttal.

---

> > > > ### Author Response · Authors · 2023-11-23
> > > > **Thank you!**
> > > >
> > > > Thank you so much for your support and engagement throughout this process! We'll do our best to incorporate all the feedback in the camera ready version.

---

### Official Review · Reviewer_tXQ4 · 2023-10-31

**Soundness:** 3 good
**Presentation:** 3 good
**Contribution:** 3 good
**Rating:** 8
**Confidence:** 3

**Summary:**

This paper proposed a conjecture about when a decoder-only autoregressive Transformer is likely to have strong length generalization ability on symbolic algorithmic tasks. There are three conditions: 1) the true next token function for the task is realizable with a *causal Transformer*; 2) the next token function can be represented by RASP-L (a subset of the RASP language) programs; 3) the training data is sufficiently diverse, which makes sure that the shortest RASP-L program can length-generalize. The conjecture states that Transformer is likely to generalize when the three conditions are satisfied. It is shown that the conjecture is supported by some case studies. On 3 tasks which have simple RASP-L programs, Transformer empirically generalize well with a moderate max train length (e.g., >30 on the *copy with unique tokens* task). On 3 tasks that "do not admit" simple RASP-L solutions, Transformers struggle to generalize. The conjecture is then applied to interpret why scratchpads can improve generalization. Specifically, on a case where the scratchpad increases the complexity of the RASP-L program, Transformers show decreased generalization performance.

**Strengths:**

- A novel approach for understanding the reasoning and out-of-distribution generalization ability of Transformers on the symbolic tasks.
- The analysis in the case studies is well supported by empirical results.
- Experiment details are well introduced.

**Weaknesses:**

- Some concepts are not clearly defined, making their meanings obscure and the statements less rigorous. For example, there is no reference or definition for the "causal Transformer" and "causal masking"; how to identify whether a RASP-L program is "simple" or "difficult".
- The correlation between RASP-L representable and the generalization of Transformer need further discussions. Are the selected cases representative enough? Is there any task which can be represented by
RASP-L but Transformers hard to generalize, or vise-versa? These answers are needed to make a stronger statement on the correlation between the two.
- No discussions on the limitations of this work.

**Questions:**

- Please refer to the weakness part.
- Minor question: Empirical results show that the max train length has a great impact on the generalization ability of Transformers. Is the proposed method helpful in analyzing this phenomenon?
- Minor suggestion: Some important definitions, e.g., that of RASP-L, would better to be included in the main paper.

---

> ### Author Response · Authors · 2023-11-15
> **Author response**
>
> Thank you for your time and effort in reviewing our paper! We are delighted that you find our approach to understanding Transformers novel and well-supported by our experiments. We are grateful that you support the acceptance of this work. We would like to address the comments and questions you’ve raised, and we’d be very open to further discussion.
>
> > Some concepts are not clearly defined, making their meanings obscure and the statements less rigorous. For example, there is no reference or definition for the "causal Transformer" and "causal masking"; how to identify whether a RASP-L program is "simple" or "difficult".
>
> Thanks for point this out. “Causal Transformers” are also often known as “decoder-only Transformers” in the literature. We have added a clarification in the updated manuscript. Essentially, with causal masking, self attention modules can only attend to tokens that came before the current token, similar to other unidirectional models.
>
> A proxy for RASP-L program simplicity is the lines of code or lines of operations in the program. We illustrate this concretely with an expanded comparison of forward vs reverse addition in Section 5.1, where we compared the lines of code in the RASP-L program for each, and showed that this corresponds to generalization performance.
>
> >The correlation between RASP-L representable and the generalization of Transformer need further discussions. Are the selected cases representative enough? Is there any task which can be represented by RASP-L but Transformers hard to generalize, or vise-versa? These answers are needed to make a stronger statement on the correlation between the two.
>
> This is a fair question. Although we cannot prove conclusively that RASP-L covers all cases of generalization success and failures, and in fact there are likely exceptions due to the complex nature of learning and generalization, we have considered most empirical analyses of Transformer length generalization on algorithmic tasks in the literature to the best of our knowledge. We have not observed exceptions in these cases. See Figure 1a for a clearer summary.
>
> Given that most studies in the literature focus on highlighting length generalization failures, we introduced a number of new tasks to showcase that Transformers do in fact have the ability to show strong length generalization. We have identified more of these positive examples than we had room for in the manuscript, such as sorting and finite state automata. We can consider adding such tasks into the full version of this paper. On the negative side, it is easy to find tasks which do not admit RASP-L programs. Dziri et al. 2023 studied a number of tasks (e.g. dynamic programming, multiplication, and certain kinds of logic puzzles), and showed failures in length generalization. We could not devise RASP-L programs for these algorithms, suggesting that they are unlikely to have simple RASP-L solutions. Nonetheless, RASP-L only covers a subset of possible programs that could be represented by a Transformer. In particular, RASP-L focuses on symbolic tasks. A large class of learnable but non-RASPable tasks are numerical algorithms, such as linear regression, which involve high-dimensional matrix and floating-point operations. We have added a discussion of the limitations in Appendix G.
>
>
> >No discussions on the limitations of this work.
>
> Thank you for pointing this out. We have added substantial discussions on limitations of this work. Please see end of Section 2 and Appendix G.
>
> >Empirical results show that the max train length has a great impact on the generalization ability of Transformers. Is the proposed method helpful in analyzing this phenomenon?
>
> In this work, we used max train length as a measure of training data diversity. As proposed by our Main Conjecture, one condition on length generalization is diversity--- the training data should be sufficiently diverse, such that there does not exist any shorter RASP-L program which agrees with the task in-distribution but not out-of-distribution. Intuitively, a dataset containing only 5 different lengths is more likely to have shortcut solutions than a dataset that contain 50 possible lengths. We have expanded Section 5 to include a deep dive on addition, which includes a section on how increasing data diversity by changing the sampling procedure also leads to improved performance. The Main Conjecture can be used to understand both phenomena (max train length and improved sampling procedures).
>
>
> **Conclusion:**
>
> Thank you for taking the time to read our rebuttal. If our response addresses your concerns, we kindly ask that you consider raising your score. Otherwise, please let us know about your remaining concerns/questions so we can make further improvements.
>
> Dziri et al: Faith and Fate: Limits of Transformers on Compositionality

---

> > ### Comment · Reviewer_tXQ4 · 2023-11-23
> > **Post rebuttal response**
> >
> > I acknowledge the authors' efforts in addressing my questions. The subsequent paper revisions and their interactive discussions with other reviewers have alleviated my initial concerns, leading to an improved score. The paper works on an important topic and proposes a viable solution, which, in my opinion, makes it suitable for being presented at the conference.

---

> > > ### Author Response · Authors · 2023-11-23
> > > **Thank you!**
> > >
> > > Thank you so much for engaging with the rebuttal process! We really appreciate your time and effort as well as your support!

---

### Official Review · Reviewer_Ho88 · 2023-11-01

**Soundness:** 2 fair
**Presentation:** 3 good
**Contribution:** 1 poor
**Rating:** 6
**Confidence:** 4

**Summary:**

This paper proposes to predict the feasibility of length generalization by checking whether the task can be implemented by RASP-L (L for Learnable).

- RASP-L is a constrained version of the RASP language, where all variables are constrained to int8. This excludes certain operations such as index arithmetic (but still allow order comparison, predecessor and successor).
- The paper shows that the difficulty in length generalization is correlated with the difficulty in writing the RASP-L solution.
    - Tasks that have a simple RASP-L solution: count, mode, copy.
    - Tasks that do not have a simple RASP-L solution: addition, parity, copy with repeated tokens.
- The paper then shows that whether scratchpad is helpful can be explained by RASP-L as well: a properly chosen scratchpad can help with performance (e.g. by making the task solvable with induction heads), but improperly chosen scratchpad, which increases the difficulty of writing in RASP-L, can hurt the performance (e.g. on the mode task).

_Post rebuttal update_: I've increased my score based on the discussions and changes in the revised paper.

**Strengths:**

- The paper analyses different factors affecting length generalization.
- The paper provides a variety of empirical evidence.

**Weaknesses:**

- The paper studies length generalization, though it's unclear what is considered as successful length generalization in the paper. [_Update_: the author has added a clarification.]
    - Fig 1(b) is considered as demonstrating successful length generalization, even though the performance from all curves are dropping, some even to 0.
    - In Fig 2(a) (i.e. the mode task), the paper seems to consider a test accuracy of 50% as reasonable generalization performance, per the comment in Sec 4.3.
- The paper proposes to use the difficulty of RASP-L programs as an indicator for length generalization performance. However, I'm not sure what this view can teach us, either theoretically or empirically. [_Update_: the authors modified Sec 5.1 significantly on implications of the RASP-L formulation.]
    - It's unclear how to empirically verify or act upon this conjecture. It's also not clear how to convert a task into RASP-L except for applying the definition directly, i.e. checking for whether certain criteria are satisfied.
        - For example, two criteria provided in the paper are 1) whether the operation is non-causal and 2) whether precise index arithmetic is required. Hence one could also say that these criteria are indicative of length generalization performance, without resorting to the RASP-L language.
    - I'm not sure whether these criteria provide new insights beyond findings in the existing literature.
        - It is well known that auto-regressive generation cannot uncover non-causal information. [_Update_: previously I misunderstood what "non-causal" mean; the authors have clarified this in the rebuttal.]
        - The concern about index arithmetic is essentially a problem with approximation (e.g. precision and weight norms), which has been raised in prior work. In particular, Hahn 2020, Chiang & Cholak 2022 and Liu et al. 2023(b) discussed limitations in the attention module, and Liu et al. 2023(a) discussed limitations in MLP.
- Some experiments have been covered in prior work.
    - The shifted position experiments are also included in Liu et al. 2023(a) (Fig 6 & 7).
    - The scratchpad experiments are similar to those in Liu et al. 2023(a). This paper uses a different form of scratchpad, but the generalization performance is not very good (e.g. generalizing from length 25 to length 50 gets around 0.3 accuracy), which is related to my earlier comment about what is considered as successful length generalization.

*References*

Hahn 2020: Theoretical Limitations of Self-Attention in Neural Sequence Models

Chiang & Cholak 2022: Overcoming a Theoretical Limitation of Self-Attention

Liu et al. 2023(a): Transformers Learn Shortcuts to Automata

Liu et al. 2023(b): Exposing Attention Glitches with Flip-Flop Language Modeling

_Note: the paper has cited Hahn 2020, Chiang & Cholak 2022 and Liu et al. 2023(a)._

**Questions:**

- Page 2, "Possible Mechanisms": it's not true that omitting positional encodings may lead to "arbitrary length" generalization, due to failures in both MLP and attention (please see the 4 citations mentioned above).
- It would be better to include discussions on prior work that relate algorithmic reasoning tasks to formal languages.
  - For example, for the discussion towards the end Section 2, please consider relating to e.g. Merrill and Sabharwal 2022 and the notion of uniformity in computational complexity (which footnote 2 has mentioned).
- Fig 3: How many test sequences are there for each trial?
- Fig 6(b): this is not really about length generalization, but more a failure of optimization?


Merrill and Sabharwal 2022: The Parallelism Tradeoff: Limitations of Log-Precision Transformers

---

> ### Author Response · Authors · 2023-11-15
> **Author response (part 1)**
>
> Thank you for your time and effort in reviewing our paper! We would like to address the comments and questions you’ve raised, and we’d be very open to further discussion. We've updated the pdf and highlighted key changes in blue.
>
> We first focus on clarifying our contribution, and its differences with prior work.
>
> > The paper proposes to use the difficulty of RASP-L programs as an indicator for length generalization performance. However, I'm not sure what this view can teach us. [...] I'm not sure whether these criteria provide new insights beyond findings in the existing literature.
>
> We emphasize that our work focuses on the empirical learnability of tasks, rather than their theoretical representability.
> While there is a long line of work on the latter question (we have included more references in Appendix G), the former question is very much still open --- it is essentially about understanding the inductive bias of Transformers.
> Our work does not resolve this question, but we believe it takes an important and significant step forward.
>
> To be concrete, we have added an explicit theoretical example (Appendix H) where our conjecture’s prediction differs significantly from prior work. We consider the popular “minimum-degree-interpolator” conjecture proposed by Abbe et al (ICML 2023 Outstanding Paper: https://openreview.net/forum?id=3dqwXb1te4).
> We give a simple theoretical setting (learning boolean conjunction) where our RASP-L conjecture correctly predicts successful length-generalization, but the min-degree conjecture does not. This simple example is emblematic of the value in our perspective.
>
> Moreover, as we discuss in the Intro (paragraph 2), the prior literature did not have a consistent answer to whether Transformers length-generalize. Most works found that standard Transformers typically fail catastrophically on even mild length-generalization, e.g. Delétang et al. 2023,   Abbe et al 2023, and Liu et al. 2023(a).  However, it was also known that Transformers can sometimes length-generalize on particular tasks. Our contribution sheds light on which tasks: what is special about these length-generalizing tasks, that separates them from non-length-generalizing ones? We claim RASP-L representation is an empirically good heuristic for answering this question.
>
> We hope these remarks help clarify the contribution of this work in the context of prior literature and address your concern regarding contribution. Please let us know if there are further concerns regarding this point.
>
> >It's unclear how to convert a task into RASP-L except for applying the definition directly, i.e. checking for whether certain criteria are satisfied. For example, two criteria provided in the paper are 1) whether the operation is non-causal and 2) whether precise index arithmetic is required. Hence one could also say that these criteria are indicative of length generalization performance, without resorting to the RASP-L language.
>
> These two criteria you mentioned are implications of our RASP-L conjecture, but our conjecture is significantly stronger.
> To give a simple example, the Parity task satisfies the two criteria you mentioned, but cannot be written in RASP-L.
>
> We acknowledge that “It's unclear how to convert a task into RASP-L” without actually trying to write the RASP-L program. This is however generally true of many computational complexity classes — for example, it's unclear whether how to determine if a task is in Polynomial time, without exhibiting a poly-time algorithm for the task. (Of course, certain well-studied complexity classes have structural features which help us determine membership. We do not yet have such a structural understanding of RASP-L, but we view the definition as an important first step). We have added a discussion of these limitations in Appendix H in the revised pdf.
>
> >It is well known that auto-regressive generation cannot uncover non-causal information.
>
> We are unaware of a formalization of this claim in the literature. This claim has been made in informal/imprecise ways, but note that it is tricky to formally define what it means for a task to “require non-causal operations” without a model of computation such as RASP-L.
>
> For example, prior works postulated that forward-order addition would not length-generalize because the naive algorithm, on a von Neumann computer, involves non-causal operations. However, Transformers empirically actually can length generalize on forward-addition, and we show there exists a RASP-L algorithm which solves this task. This algorithm is not the naive one; it is specific to the Transformer architecture (not the von Neumann architecture), and it is thus able to “work around” the causality restrictions by taking clever advantage of parallelism.

---

> ### Author Response · Authors · 2023-11-15
> **Author response (part 2)**
>
> >Some experiments have been covered in prior work. The shifted position experiments are also included in Liu et al. 2023(a) [...] The scratchpad experiments are similar to those in Liu et al. 2023(a).
>
> We first emphasize that the primary contribution of our work is not in the experiments, but in the conceptual framework which captures the results of those experiments (the RASP-L definition and conjecture). Indeed, our goal is to help understand the various experimental phenomena that have been observed in the past. Many papers, including Liu et al. 2023(a), have noticed the benefits of scratchpads — these papers motivate our work.
>
> As a technical detail, note that the shifted-position experiments are not identical to Liu et al. 2023(a). Our paper does not propose a new shifting scheme — it simple replicates the standard practice in large-scale LLMs of training on random intervals from an iid stream of examples, and using learnt positional encodings for the entire context (e.g. Karpathy, 2023; Brown et al., 2020). Nevertheless, we agree that there is a structural similarity, so we have added a note about this in Section 3, Footnote 5.
>
> >it’s not true that omitting positional encodings may lead to "arbitrary length" generalization, due to failures in both MLP and attention
>
> Thank you for flagging this; we will clarify the statement in the paper. Note that the issue is subtle: there exist certain tasks where arbitrary-length generalization is provably possible, and such failures of MLP and attention do not manifest. For example, consider the simple Transformer that solves the boolean conjunction task in Appendix H. (The limitations of Hahn 2020 do not apply in our setting, since we allow weights to take values in the extended real line, as described in Footnote 3).
>
> >It would be better to include discussions on prior work that relate algorithmic reasoning tasks to formal languages.
>
> We have added more discussion of such works in Appendix G, and we also have added a footnote about uniformity to Section 2 as suggested.
>
> >The paper studies length generalization, though it's unclear what is considered as successful length generalization in the paper.
>
> Thank you for this question. In this paper, we consider successful length generalization to mean near perfect test EM on length that is at least 10 longer than the maximum training length, and we consider failures of length generalization to mean near 0 test performance on +10 length. We have added this clarification in a new figure (see Fig 1a) that summarizes the different tasks.
>
> The specific choice of 10 is somewhat arbitrary. It reflects the level that all tasks which demonstrates non-trivial generalization can readily achieve without significant changes to model / data size and training strategy. Certain tasks such as count also exhibit significantly stronger length generalization than +10. However, it is also a reasonable proxy for capturing the differences between the set of tasks that are amenable to RASP-L solutions and the set that is not. Existing literature on length generalization reports many failure cases, including addition and parity, and in these cases performance starts to degrade as soon as the length goes out-of-distribution (see e.g. Nye et al Figure 3 and Anil et al). Being able to generalize perfectly for length of +10 or more suggests a qualitative difference from the failures cases reported in the literature. Nevertheless, how to predict the exact level of length generalization for a given task is still an open question.
>
> >Fig 3: How many test sequences are there for each trial?
>
> The test data size is 5 * batch size for each task, which is typically 5 * 128 = 640 examples.
>
> >Fig 6(b): this is not really about length generalization, but more a failure of optimization?
>
> Indeed, for parity without scratchpad we see the difficulty of optimization, which has also been noted in prior work (e.g. Chiang & Cholak and Bhattamishra et al). We focus on comparing parity with different scratchpads, based on how simple the scratchpads are according to RASP-L. We observe that the tasks with simpler RASP-L solutions also show faster optimization, which lends some support to the intuition that simple RASP-L programs are “easier-to-learn”.
>
> **Conclusion:**
>
> Thank you for taking the time to read our rebuttal. If our response addresses your concerns, we kindly ask that you consider raising your score. Otherwise, please let us know about your remaining concerns/questions so we can make further improvements.

---

> > ### Author Response · Authors · 2023-11-15
> > **Additional references**
> >
> > Nye et al: Show Your Work: Scratchpads for Intermediate Computation with Language Models
> >
> > Anil et al: Exploring Length Generalization in Large Language Models
> >
> > Chiang & Cholak: Overcoming a Theoretical Limitation of Self-Attention
> >
> > Bhattamishra et al: On the Ability and Limitations of Transformers to Recognize Formal Languages

---

> > > ### Comment · Reviewer_Ho88 · 2023-11-19
> > > **Further clarifications**
> > >
> > > Thank you for the detailed responses and clarifications! My main concern was that the main claim (i.e. the RASP-L conjecture) is stated stronger and more general than what the empirical results would support.
> > >
> > > The authors have made several updates to the draft which I appreciate, including:
> > >   - a more detailed description on the experimental setups;
> > >   - an implication of RASP-L (Sec 5.1);
> > >   - discussions on the subtlety of various claims which I found were previously too strong, as well as limitations and comparison to prior work.
> > >
> > > I have a few more questions that I'd like to have the authors' thoughts on:
> > > - Length generalization can be considered as a specific type of OOD generalization. Is RASP-L informative for OOD generalization in general, or is it specifically for length generalization?
> > >   - A related work is Liu et al. 2023(b), which studies challenges in Transformer's OOD generalization under the same length. Could you please comment on the comparison?
> > > - For conjunction, what's the performance if we train on the training distribution in H1, but test on more numbers of 0s? Or is this type of generalization not supported by a short RASP-L program?
> > > - Maybe I missed this, but are there counterexamples, where the task has a short RASP-L program but cannot length generalize? Or, how to prove that there are no such tasks?
> > >   - For example, FIRST (i.e. whether the first token in a binary string is 1; please see Chiang & Cholak 22) can be implemented with 1 line of RASP-L (is this correct?), but 1) can length generalize with layernorm (eps=0) and 2) cannot length generalize without layernorm or with layernorm but eps>0. It seems that RASP-L (or RASP in general) is not sensitive to architecture details like this, which actually impacts the practical performance of Transformers. Could you comment on this please?
> > > - I'd like to have a clearer understanding on the applicability of RASP-L: the authors added discussions in Appendix G on the limitations of RASP-L, such as not able to represent numerical algorithms. Could you please comment on what is the set of tasks where RASP-L can determine length generalization abilities?

---

> > > > ### Author Response · Authors · 2023-11-20
> > > > **Author response (part 1)**
> > > >
> > > > Thank you for your response! We are grateful for your continued engagement.
> > > >
> > > > >For conjunction, what's the performance if we train on the training distribution in H1, but test on more numbers of 0s? Or is this type of generalization not supported by a short RASP-L program?
> > > >
> > > > This is an interesting question! We ran some experiments using the same train distribution, but the test distribution now has a random number of 0s between 0 and 5 (instead of 0 or 1 in the training set). We train models on length 20. On test sequences of length 20 (and 0s up to 5), we find 100% test accuracy for the median over 20 runs (in fact for all runs). Further, we find that this task length generalizes also (100% accuracy on sequences of length 50).
> > > >
> > > > >Length generalization can be considered as a specific type of OOD generalization. Is RASP-L informative for OOD generalization in general, or is it specifically for length generalization?
> > > > >A related work is Liu et al. 2023(b), which studies challenges in Transformer's OOD generalization under the same length. Could you please comment on the comparison?
> > > >
> > > > This is a great question. We do indeed think that the intuitions behind the RASP-L conjecture can be applicable to compositional generalization more broadly (as we allude to in the Conclusion), though there may be different considerations regarding the forms of data diversity required for different types of compositional generalization. We believe that this is a natural extension in future work.
> > > >
> > > > For now, we have some preliminary support for this. One is seen in the OOD example on more 0s that you suggested for the conjunction task. We find that the model generalizes easily in this setting. Another example is a new systematic split we tried for the addition task. We train models on a distribution where the last digits of the numbers are always even during training, and are odd at test time. We observe that generalization performance in this setting is similar to those presented in the paper, that is we observe perfect generalization to test split of the same length, and similar length generalization performance on longer lengths as the experiments in the paper. We will include these new results and discussion in the camera-ready version of the paper.
> > > >
> > > > _Re. Liu et al. 2023(b):_
> > > >
> > > > This is an interesting question. While we do not make any claims about this kind of generalization in our paper, we can try to apply the RASP-L perspective regardless. The FFLM task does have a simple RASP-L solution. Apparently, from the experiments in Liu et al. 2023(b), this solution is not always learnt in a fully robust way. However, we find it interesting that the OOD generalization is still far from random chance (e.g. between 1e-1 and 1e-3 in Figure 3, bolded box) — suggesting that the model has learned something perhaps “close” to the correct solution. Finally, we speculate that adding more forms of “data diversity” could further improve generalization (as we found in our paper). For example, if the model is trained on FFLM tasks with p={all multiples of 5%}, then it is possible the model will generalize to many other intermediate values of p.
> > > >
> > > > >Maybe I missed this, but are there counterexamples, where the task has a short RASP-L program but cannot length generalize? Or, how to prove that there are no such tasks?
> > > >
> > > > We do not know of any counterexamples in that direction — i.e. tasks with short RASP-L programs but poor length-generalization. This was by design; we were careful to define RASP-L to forbid the known counterexamples in original RASP.
> > > >
> > > > Nevertheless, this statement is currently a conjecture. Moreover, we do not expect a proof of this to be easy using current techniques: our statement is a claim about learning (not representation) in Transformers trained with SGD/Adam, and the current theoretical understanding of Transformer learning is in its infancy (most known results are only for single-layer Transformers on simple tasks, for example). We hope our empirical conjecture can guide future theoretical results.

---

> > > > > ### Author Response · Authors · 2023-11-20
> > > > > **Author response (part 2)**
> > > > >
> > > > > >For example, FIRST (i.e. whether the first token in a binary string is 1; please see Chiang & Cholak 22) can be implemented with 1 line of RASP-L (is this correct?), but 1) can length generalize with layernorm (eps=0) and 2) cannot length generalize without layernorm or with layernorm but eps>0. It seems that RASP-L (or RASP in general) is not sensitive to architecture details like this, which actually impacts the practical performance of Transformers. Could you comment on this please?
> > > > >
> > > > > It is correct that FIRST can be implemented in 1 line of RASP-L. However, it is also true that FIRST empirically length-generalizes to a significant degree (even stronger than most tasks we consider), and so this is consistent with our conjecture. Empirically, note that Chiang & Cholak 22 observe degradation of length-generalization only at extremely large OOD lengths. E.g. in Figure 4 of CC22, they train on length n=10 and test on length n=1000, and observe poor generalization. However, in that same figure, they report that training on length n=300 (and testing on n=1000) performs nearly perfectly (top left, purple line) — and performance continues to improve with more epochs. Thus, we consider our results to be consistent with Chiang & Cholak 22; our different interpretations are due to differences in what we consider “good length generalization.” Still, we acknowledge that our work does not fully specify the extent/degree of length generalization which can be expected, and we do not study the full effects of various hyperparameter choices, which we further discussed in the new Limitations section.
> > > > >
> > > > > >I'd like to have a clearer understanding on the applicability of RASP-L: the authors added discussions in Appendix G on the limitations of RASP-L, such as not able to represent numerical algorithms. Could you please comment on what is the set of tasks where RASP-L can determine length generalization abilities?
> > > > >
> > > > > We expect RASP-L to apply on “symbolic” algorithmic tasks. One way to formally define this is: tasks which can be solved exactly in the “word-RAM” model of computation. Informally, this can be thought of as “any algorithm that can be written in C without floating-point operations.” Such tasks are sometimes called “elementary algorithms” (e.g. Liu et al. 2023a), and they are the focus of many existing studies on algorithmic generalization of neural networks. To be concrete, symbolic tasks include almost all of the algorithms in CLRS, except for randomized algorithms and linear-programming.
> > > > >
> > > > > Thanks again for your discussion, we think that it has greatly improved our paper. We hope you will support the paper’s acceptance after these improvements!
> > > > >
> > > > > CLRS: Introduction to Algorithms, Thomas H. Cormen, Charles E. Leiserson, Ronald L. Rivest, and Clifford Stein

---

> ### Comment · Reviewer_Ho88 · 2023-11-22
>
> Thank you for the further discussions!
>
> - Re Liu et al. 23(b): I agree that improving data diversity will likely help, which Liu et al. 23(b) also pointed out (their R4) as well as Jelassi et al. 23 ("priming"). Note though this might be considered as a "cheating" solution, since introducing more data reduces the OOD problem into an in-distribution problem.
>
> - Chiang & Cholak 21's results are based on aggressive length generalization: I agree, and I think this again showcases the ambiguity in the definition of "length generalization". I hope this definition could be further highlighted in the paper since it will better clarify the implication/scope of the results; currently it seems that the only place that mentioned the "generalizing to 10 more length" is the caption of Figure 1 which I think is easy to miss.
>
> - Re counterexamples: It would be great if the authors could include this comment in the limitation or the conclusion as well.
>
> Given these changes, I'm happy to raise my score to lean on the side of acceptance. The main reason for my rating is that I think the paper provides an interesting concept, and I hope it could be better tested out and the paper should discuss the limitations and relations to prior work more carefully.

---

> > ### Author Response · Authors · 2023-11-22
> > **Thank you!**
> >
> > Thank you so much for your support and engagement! We will include these final discussions in the camera ready version.

---

### Author Response · Authors · 2023-11-15
**Comment to All Reviewers**

We thank all reviewers for their thoughtful comments. To address these remarks, we have made several changes to the pdf, highlighted in blue. Our individual responses to reviewers will be posted separately, in addition to this common response.

Summary of major changes:

* Added Appendix Section I (pg 26): To illustrate how our conjecture differs from the existing literature, we give a direct comparison to the “min-degree interpolator” conjecture of Abbe et al (ICML 2023 Outstanding Paper: https://openreview.net/forum?id=3dqwXb1te4). Specifically, we show a simple theoretical setting where the conjecture of Abbe et al. does not correctly predict Transformers’ out-of-distribution behavior, but our RASP conjecture does.
* Figure 1 is updated with a summary of experimental results, showing the distinguishing power of RASP-L. Successful length generalization here is defined as near perfect test performance on length +10 more than max training length.
* Added discussion of limitations following the Main Conjecture (end of Section 2). Added a full discussion on limitations within the context of additional related works in Appendix H.
* Re-organized Section 5 with a fuller analysis of the addition task, showing how RASP-L program complexity (measured by lines of code) is correlated with likelihood of length generalization, and demonstrating the two levers we have to improve generalization performance: 1. reduce RASP-L complexity, 2. increase data diversity.

---

### Author Response · Authors · 2023-11-22
**Gentle reminder**

Dear reviewers,

Thank you again for your time and effort in helping review this work! This is a gentle reminder that the discussion period closes tomorrow, and we kindly ask you to let us know if there are any remaining concerns or questions, and to consider updating your scores if we have been able to address your questions.

Regards,
Authors

---

### Meta-Review · Area_Chair_og5c · 2023-12-09

**Metareview:**

This paper investigates when and how decoder-only transformers are able to generalize over length, despite some empirical indication that learning to reason in simple reasoning scenarios is hard. To this end, authors propose a conjecture about when such transformers can have strong length generalization ability on symbolic algorithmic tasks. This conjecture relies on the formalism of the RASP-L programming language and partial empirical evidence is provided to support this conjecture.

Reviewers appreciated the proposed conjecture and the perspective on generalization that the paper is bringing. However, some also raised several concerns regarding presentation, the rigor of the formalization and reason how the experiments are supporting the conjecture. During the rebuttal the authors managed to address many concerns and flipped two scores towards full acceptance.

The paper is accepted as while the conjecture might not hold in practice, the bridge with the RASP language is interesting and can provide significant insights to built novel probes into what transformers are able to learn and under which conditions guaranteed to generalize.

**Justification For Why Not Higher Score:**

The contribution, while touching the important task of generalization in modern deep learning models, is limited in scope to certain transformers architectures and tasks.

**Justification For Why Not Lower Score:**

The paper brings a novel perspective on measuring generalization via a program induction reduction and can be of interest to the ICLR community.

---

### Decision · Program_Chairs · 2024-01-16

Accept (poster)